# Vulnerability of drug-resistant EML4-ALK rearranged lung cancer to transcriptional inhibition

Athanasios R Paliouras[1,2,†], Marta Buzzetti[1,3,†], Lei Shi[1,2,†], Ian J Donaldson[4], Peter Magee[1,2], Sudhakar Sahoo[5], Hui-Sun Leong[5], Matteo Fassan[6], Matthew Carter[2,7], Gianpiero Di Leva[8], Matthew G Krebs[2,7], Fiona Blackhall[2,7], Christine M Lovly[9] & Michela Garofalo[1,2,*]

## Abstract

A subset of lung adenocarcinomas is driven by the EML4-ALK translocation. Even though ALK inhibitors in the clinic lead to excellent initial responses, acquired resistance to these inhibitors due to on-target mutations or parallel pathway alterations is a major clinical challenge. Exploring these mechanisms of resistance, we found that EML4-ALK cells parental or resistant to crizotinib, ceritinib or alectinib are remarkably sensitive to inhibition of CDK7/12 with THZ1 and CDK9 with alvocidib or dinaciclib. These compounds robustly induce apoptosis through transcriptional inhibition and downregulation of anti-apoptotic genes. Importantly, alvocidib reduced tumour progression in xenograft mouse models. In summary, our study takes advantage of the transcriptional addiction hypothesis to propose a new treatment strategy for a subset of patients with acquired resistance to first-, second- and third-generation ALK inhibitors.

**Keywords** ALK/EML4 translocation; ALKi; CDKi; drug resistance; NSCLC
**Subject Category** Cancer

## Introduction

In non–small-cell lung cancer (NSCLC), small molecule inhibitors that target mutant kinases have offered unprecedented success in the management of the disease. One of the most successful examples is echinoderm microtubule like-4-anaplastic lymphoma kinase (EML4-ALK)-mutant lung cancer, which affects 4–5% of lung cancer patients (Gainor *et al*, 2013). A fusion of ALK with EML4 (Soda *et al*, 2007) causes the constitutive activation of the ALK kinase domain and subsequent oncogenic signalling, typically through the MAPK, JAK-STAT and PI3K-AKT pathways (Chiarle *et al*, 2008). To date, the first-generation ALK inhibitor crizotinib, second-generation ALK inhibitors ceritinib, alectinib and brigatinib, and the third-generation ALK inhibitor lorlatinib have been approved by the Food and Drug Administration (FDA) for the treatment of patients with lung cancer harbouring the EML4-ALK translocation. The objective response rate for the ALK inhibitors crizotinib and alectinib in the clinic surpasses 60%, while a median progression-free survival of 34 months has been demonstrated with alectinib (Camidge *et al*, 2018). However, patients eventually develop disease progression due to drug resistance.

A common mechanism of resistance is represented by mutations in the ALK kinase domain that hinder small molecule binding, such as the G1202R mutation which occurs after alectinib treatment (Gainor *et al*, 2016). A multitude of parallel pathways' alterations can also cause resistance to ALK inhibitors, compensating for the lack of EML4-ALK activity. Usually, these are receptor tyrosine kinases (RTKs), such as EGFR, HER3/4 and c-KIT(Lin *et al*, 2017) but they can also be other oncogenes, such as KRAS (Doebele *et al*, 2012). Treatment options upon failure of ALK inhibitors are limited and chemotherapy offers only a short-lived benefit to these patients whereas the additive benefit of immunotherapy in this context is still unclear (Pacheco & Camidge, 2019).

In this study, we discovered a cell cycle dysregulation and a vulnerability of EML4-ALK lung cancer cells to the pan-CDK inhibitors alvocidib and dinaciclib, as well as the CDK7/12 inhibitor THZ1. We put forward the idea of testing these inhibitors in the clinic after ALK inhibitors have failed due to the development of acquired resistance.

1  Transcriptional Networks in Lung Cancer Group, Cancer Research UK Manchester Institute, University of Manchester, Manchester, UK
2  Cancer Research UK Lung Cancer Centre of Excellence, Manchester and University College London, London, UK
3  Biomedical Research Centre, School of Science, Engineering and Environment, University of Salford, Salford, UK
4  Bioinformatics Core Facility, Faculty of Biology, Medicine and Health, University of Manchester, Manchester, UK
5  Computational Biology Support, Cancer Research UK Manchester Institute, University of Manchester, Manchester, UK
6  Department of Medicine, Surgical Pathology & Cytopathology Unit, University of Padua, Padua, Italy
7  Division of Cancer Sciences, Faculty of Biology, Medicine and Health, Christie Hospital, University of Manchester, Manchester, UK
8  School of Pharmacy and Bioengineering, Guy Hilton Research Institute, Keele University, Keele, UK
9  Vanderbilt-Ingram Cancer Center, Vanderbilt University Medical Center, Nashville, TN, USA
   *Corresponding author. Tel: +44 161 3060838; E-mail michela.garofalo@manchester.ac.uk
   †These authors contributed equally to this work

# Results

### Pathways dysregulated in crizotinib resistance

To mimic the context of acquired resistance to ALK inhibitors *in vitro*, we employed cell lines with acquired resistance to crizotinib (Fig 1A and B), ceritinib and alectinib (Appendix Fig S1A) obtained by long-term exposure to increasing concentrations of the drugs. These cell lines were derived from the parental H3122 cells (Lovly *et al*, 2011) and STE-1 cells, a patient-derived lung cancer cell line described in (Lovly *et al*, 2014), both of which carry the EML4-ALK (E13;A20) translocation.

As revealed by Sanger sequencing, all the resistant cells have wild-type ALK kinase domain. Due to the absence of ALK kinase domain mutations, we reasoned that this lack of response to ALK inhibition was a result of alterations in parallel signalling. To identify the driver of crizotinib resistance, we followed a transcriptomic approach and performed RNA-seq of CrizR1 and isogenic parental H3122 cells (Dataset EV1). Through this analysis, we detected an upregulation of the *EGFR* mRNA that we further validated at the protein level (Appendix Fig S1B). Increased EGFR signalling, through ligand upregulation, gene amplification or point mutation, is to our knowledge the most common ALK-independent mechanism of resistance to ALK inhibitors (Camidge *et al*, 2014). To investigate whether this was the main driver of resistance, we silenced EGFR by RNAi (Appendix Fig S1C). We asked if this silencing in combination with crizotinib could re-sensitize CrizR1 cells; however, we observed no significant induction of apoptosis (Appendix Fig S1D). We used HCC-827 EGFR-driven cells as positive control, which indeed became apoptotic upon EGFR silencing. In addition, we detected an upregulation of the TGF-β receptors 1 and 2 (Appendix Fig S1E) and found that inhibition of TGF-β activity with the small molecule inhibitor galunisertib resulted only in a marginal decrease of cell proliferation (Appendix Fig S1F). Ruling out that EGFR/TGF-βR act as drivers of resistance, we searched for more dysregulated oncogenes.

Using the HALLMARK gene collection, we found a significant enrichment in epithelial-to-mesenchymal transition (EMT)-related genes in CrizR1 cells (Fig 1C and D). Given the mesenchymal phenotype of the crizotinib-resistant cell lines (Appendix Fig S2A), we asked whether EMT played a role in these drug-resistant cells. We chose 4 genes known to induce mesenchymal characteristics and confirmed their expression levels via qPCR. *AXL, LOX, SNAI2* and *VIM* were upregulated in the majority of the resistant cell lines (Fig 1E). AXL protein levels were particularly elevated in the CrizR1 and CrizR4 cells and AXL is known to be activated in drug-resistant EML4-ALK cells (Nakamichi *et al*, 2018). Interestingly, we detected a downregulation of ALK in the same cells, raising the possibility that AXL compensates in part for the reduced EML4-ALK activity. Moreover, AXL inhibition with the small molecule inhibitor bemcentinib (Holland *et al*, 2010) resulted in the downregulation of *LOX, SNAI2* and *VIM,* indicating that AXL activation is responsible for the induction of these genes and subsequently EMT (Fig 1F). Next, we asked whether AXL upregulation is functional in these cells and in a proliferation assay, bemcentinib halted proliferation in CrizR1 and CrizR4 cells in combination with crizotinib (Fig 1G), suggesting that AXL activation has a functional role in these cells. However, bemcentinib alone or in combination with crizotinib did not induce cell death or senescence (Fig 1H and Appendix Fig S2B), indicating a cytostatic instead of a cytotoxic effect. In summary, we have detected an AXL-mediated induction of resistance to crizotinib. Although AXL inhibitors significantly reduce cell proliferation, they are unable to kill crizotinib-resistant cells.

### Dysregulation of cell cycle-related genes in crizotinib-resistant cells

In the RNA-seq data comparing crizotinib-resistant versus crizotinib-sensitive cells, a KEGG pathway analysis by GSEA revealed 9 pathways enriched in dysregulated genes (Dataset EV1 and Fig 2A). Among them, there was a significant enrichment in cell cycle-related genes (Fig 2A and B, Dataset EV2). We were able to confirm by immunoblot the upregulation of multiple cell cycle-related genes in the crizotinib-resistant cells. Notably, CDK1 and CCNB1, as well as CDK6, were upregulated in the majority of the resistant cell lines (Fig 2C). CDK2 was not upregulated, but we found an upregulation

---

**Figure 1. EMT-related genes are dysregulated in crizotinib-resistant cells.**

A Table reporting all drug-resistant cell lines used in this study.

B Proliferation assay of H3122 parental and isogenic drug-resistant cells, treated with the indicated concentrations of crizotinib for 72 h. $P < 0.0001$ was calculated for IC$_{50}$ shift as indicated in the Materials and Methods.

C Gene set enrichment analysis after RNA-seq of H3122 sensitive versus CrizR1 crizotinib-resistant cells.

D Epithelial-to-mesenchymal transition enrichment plot from the RNA-seq of C), using the HALLMARK gene collection.

E H3122 parental and drug-resistant isogenic cell lines were RNA-extracted, and gene expression was analysed by RT-qPCR. On the right hand side is a Western blot analysis of the AXL and EML4-ALK protein levels from the same cell lines. AXL: CrizR1 versus H3122 $P < 0.001$, CrizR4 versus H3122 $P < 0.001$, CrizR5 versus H3122 $P = 0.03$, CeritR versus H3122 $P = 0.2$; LOX: CrizR1 versus H3122 $P < 0.001$, CrizR4 versus H3122 $P < 0.001$, CrizR5 versus H3122 $P < 0.001$, CeritR versus H3122 $P = 0.008$; SNAI2: CrizR1 versus H3122 $P < 0.001$, CrizR4 versus H3122 $P < 0.001$, CrizR5 versus H3122 $P = 0.003$, CeritR versus H3122 $P = 0.003$; VIM: CrizR1 versus H3122 $P < 0.001$, CrizR4 versus H3122 $P < 0.001$, CrizR5 versus H3122 $P < 0.001$, CeritR versus H3122 $P < 0.001$.

F Gene expression analysis by RT-qPCR of CrizR1 cells treated with the indicated concentrations of bemcentinib for 48 h. LOX: $P = 0.03$; SNAI2: $P = 0.02$; VIM: $P = 0.03$.

G Proliferation assay of CrizR1 and CrizR4 cells treated with the indicated concentrations of bemcentinib $\pm$ 1 μM crizotinib for 72 h. ***$P < 0.001$ for IC$_{50}$ shift, as indicated in the Materials and Methods ($n = 4$).

H Annexin V+ apoptotic assay in CrizR1 cells treated with 1 μM crizotinib, 2.5 μM bemcentinib, or combination. STE-1 parental cells were used as crizotinib drug control. crizotinib: $P = 0.2$; bemcentinib: $P = 0.4$; combination: $P = 0.3$.

Data information: Statistical comparisons were performed using a paired, two-tailed Student $t$-test. Plotted graphs show mean $\pm$ SD ($n = 3$, unless otherwise specified).
*$P < 0.05$, ***$P < 0.001$, N.S. = Not Significant $P > 0.05$.
Source data are available online for this figure.

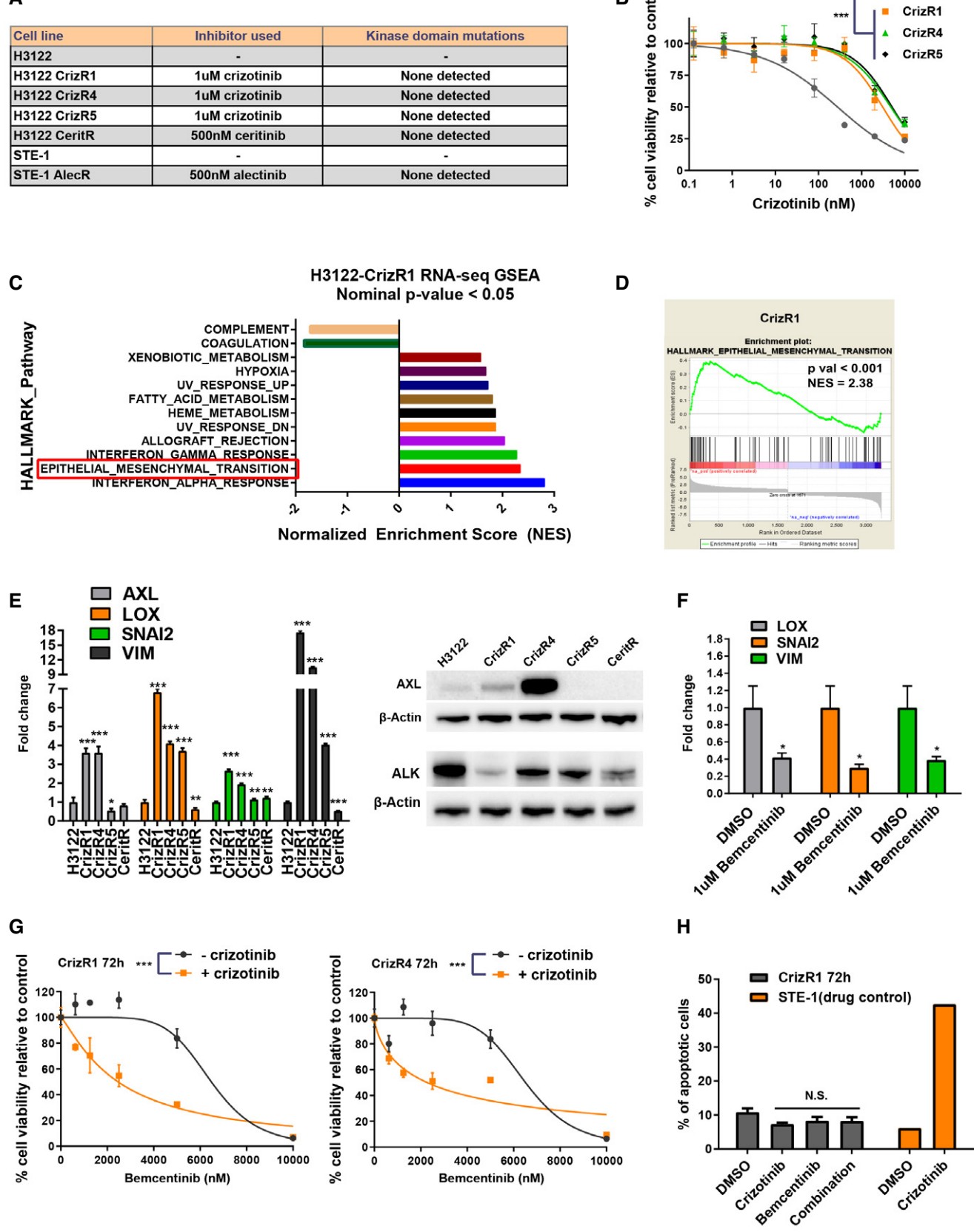

Figure 1.

of its partner CCNE1. In alectinib-resistant cells, CDK1, CCNB1 and CDK6 were also upregulated (Fig 2D).

With CDK6 being the most strongly upregulated protein, we hypothesized that pharmacological inhibition of CDK6 would reverse the resistance to crizotinib. To this end, we used the specific CDK4/6 inhibitor palbociclib in a proliferation assay and identified a limited sensitization to crizotinib only at high micromolar concentration (Fig EV1A). We reasoned that upregulation of CDK6 on its own might not be sufficient to induce crizotinib resistance. Therefore, we used the CDK inhibitor alvocidib (Flavopiridol), a potent inhibitor of CDK1, CDK2, CDK4, CDK6 and CDK9, with or without crizotinib in a proliferation assay. Although alvocidib did not synergize with crizotinib in CrizR1 cells (Fig 2E and F), we observed a remarkable sensitivity to single-agent alvocidib in both parental and isogenic resistant cells (Fig 2G). In addition to the H3122, which carry the EML4-ALK variant 1, we also tested H2228 cells, which harbour the EML4-ALK v3a/b variant and are primarily more resistant to crizotinib compared with H3122 cells. Alvocidib treatment resulted in comparable inhibition of proliferation (Fig EV1B).

To follow up on this observation, we also tested dinaciclib, a newer, more potent CDK1, CDK2, CDK5 and CDK9 inhibitor (Parry et al, 2010). Both alvocidib and dinaciclib significantly inhibited cell proliferation at low nanomolar concentrations in EML4-ALK cells with acquired resistance to these inhibitors as well as the parental cells (Fig 2H).

We then asked whether CDK1, CDK2 or CDK6 could affect the resistance to crizotinib individually or in synergy. Upon silencing of these CDKs (Fig EV1C) and after a cell cycle profile, it was evident that CDK6 silencing did not affect these cells, while CDK1 silencing (or silencing of all the three CDKs simultaneously) resulted in arrest of the cell cycle in the G2/M phase and an induction of cell death as assessed by DNA content (Fig EV1D and E).

## CDK inhibitors induce apoptosis through transcriptional inhibition

Considering the efficacy of both alvocidib and dinaciclib in inhibiting the transcriptional CDKs, and the preferential activity against CDK9, we next hypothesized that their pronounced anti-proliferative activity could be in part due to the suppression of transcription. To evaluate the transcriptional hypothesis, we used the CDK7/12 inhibitor THZ1 (Kwiatkowski et al, 2014), since CDK7 is also

influencing transcription by phosphorylating the RNA polymerase II (Blagosklonny, 2004). As with alvocidib, we followed up on the effectiveness of THZ1 as a single agent. All EML4-ALK cells showed a remarkable decrease in cell proliferation upon THZ1 treatment (Fig 3A).

Treatment of the parental, crizotinib-, ceritinib- or alectinib-resistant cells with modest concentrations of alvocidib, dinaciclib or THZ1 led to near-complete inhibition of cell proliferation as assessed by crystal violet staining (Fig 3B). These data suggest that CDK inhibition may target an inherent vulnerability of EML4-ALK lung cancer and should be further tested.

We next questioned whether this pronounced effect observed upon treatment with CDK inhibitors was a result of cell cycle arrest or apoptotic cell death. Alvocidib treatment caused an accumulation of cells in the G2/M phase assessed by DNA content (Fig EV2A). Apoptotic cell death was the main outcome of alvocidib and dinaciclib treatment in all cell lines tested, assessed via PARP cleavage (Fig EV2B) as well as Annexin V+/PI staining (Figs 3C and EV2C). In addition, this induction of apoptosis was not due to toxicity, as normal lung epithelial HBEC cells did not become apoptotic (Fig EV3A). Furthermore, there was selectivity towards EML4-ALK cells, as alvocidib and dinaciclib induced significantly higher levels of apoptosis in CrizR1 and AlecR cells compared with KRAS$^{mut}$ A549 cells, KRAS$^{wt}$/EGFR$^{mut}$ PC-9 cells or cells isolated from metastatic sites (H1299 and H460) (Figs 4A and EV3B).

To approach the induction of apoptosis in a more unbiased fashion, we treated CrizR1 cells with alvocidib and used an apoptosis array that profiled 43 different proteins (Figs 4B and EV3C). We found an upregulation of critical pro-apoptotic components such as BIM, BID, SMAC and as expected, CASP3 and CASP8. In agreement with a previous study (Ma et al, 2003) after alvocidib or dinaciclib treatment, we observed a downregulation of MCL-1 as well as of the anti-apoptotic protein BIRC5 (Survivin) in all EML4-ALK cells resistant to all ALK inhibitors (Fig 4C).

We then reasoned that all the three inhibitors affect transcriptional regulation and subsequently the mRNA levels of pro- and anti-apoptotic proteins. Alvocidib is known to decrease transcriptional output by inhibiting CDK9 and, consequently, elongation by the RNA Polymerase II (Blagosklonny, 2004). Indeed, in our system, alvocidib and dinaciclib treatment decreased phosphorylation at the Ser2 repeat of the RNA Pol II (Fig EV3D). To test the specificity of these inhibitors, we used siRNAs for CDK7 and CDK9. Only a modest knockdown of CDK9 was sufficient to induce apoptosis

---

**Figure 2. Actionable cell cycle dysregulation in crizotinib-resistant cells.**

A    GSEA enrichment analysis using the KEGG gene set identifiers. Shown are the significantly dysregulated pathways ($P < 0.05$).

B    Cell cycle enrichment plot from (A).

C    Western blot analysis of H3122 parental and isogenic drug-resistant cell lines for the indicated proteins.

D    Western blot analysis of Ste-1 parental and isogenic alectinib-resistant cell lines for the indicated proteins.

E    Proliferation assay of CrizR1 cells treated with the indicated concentrations of alvocidib $\pm$ 1 μM crizotinib for 72 h. $P = 0.2$ ($n = 4$).

F    As above, in a proliferation assay, H3122 and CrizR1 cells were treated in parallel with DMSO or 1 μM crizotinib as drug control. H3122 versus DMSO $P < 0.0001$, CrizR1 versus DMSO $P = 0.1$ ($n = 4$).

G, H    Proliferation assay of CrizR1, CrizR4, CrizR5, CeritR and AlecR isogenic drug-resistant cell lines, treated with the indicated concentrations of alvocidib (G) or dinaciclib (H) for 72 h. $P > 0.05$ was calculated for IC$_{50}$ shift, as indicated in the Materials and Methods.

Data information: Statistical comparisons were performed using a paired, two-tailed Student t-test. Plotted graphs show mean $\pm$ SD ($n = 3$, unless otherwise specified).
***$P < 0.001$, N.S. = Not Significant $P > 0.05$.
Source data are available online for this figure.

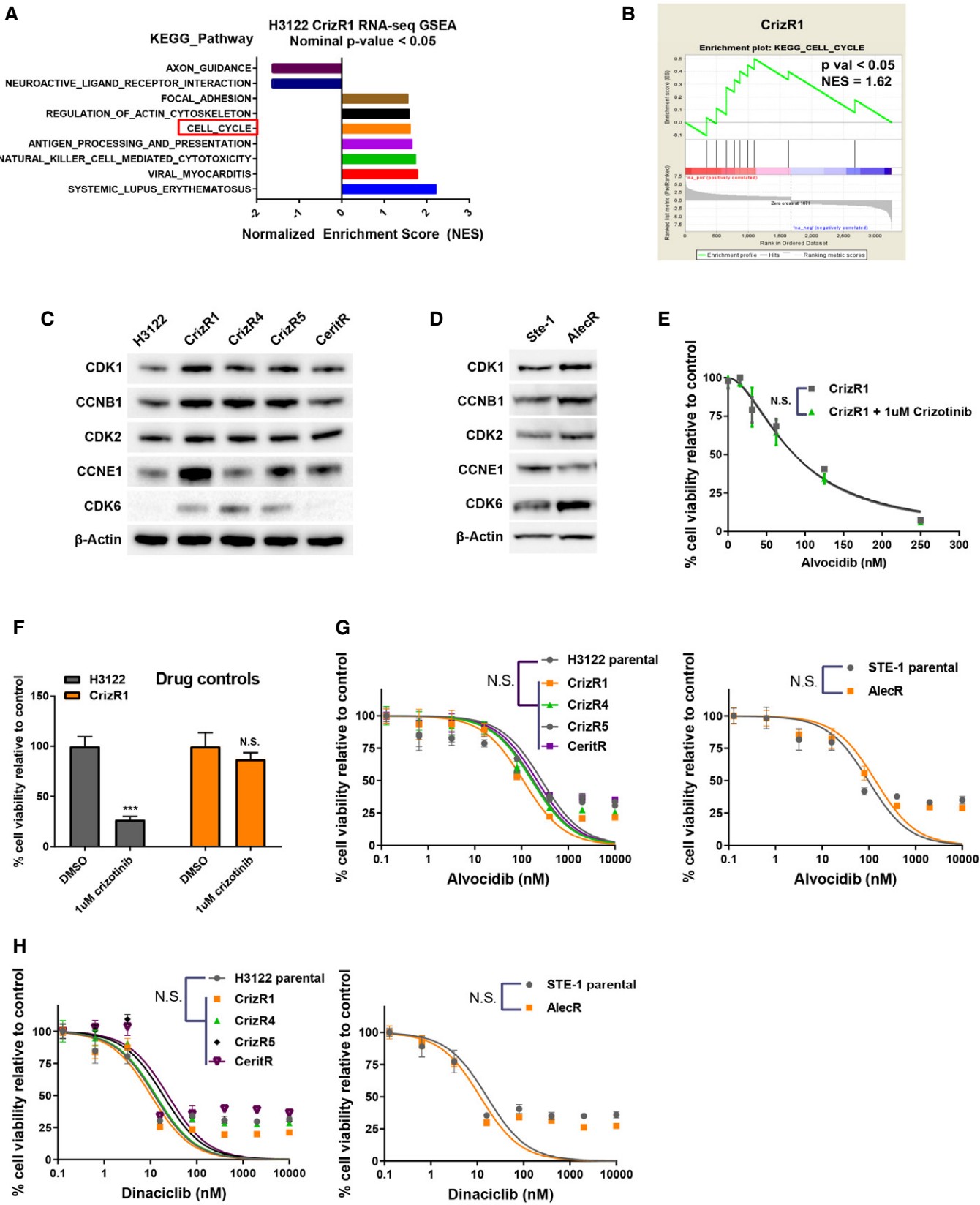

Figure 2.

**A**

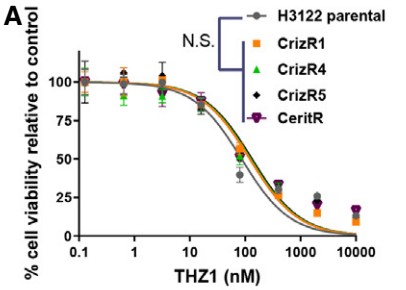

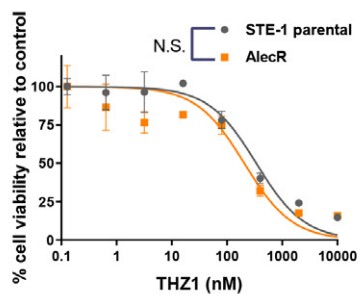

**B**

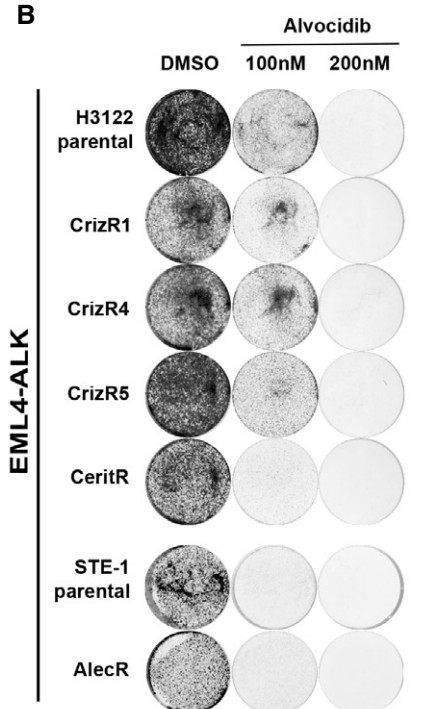

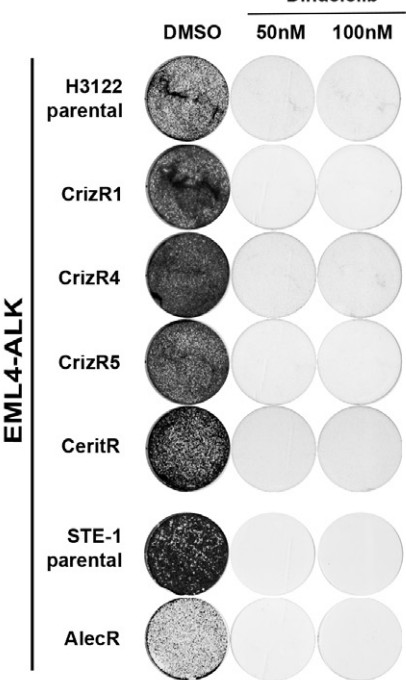

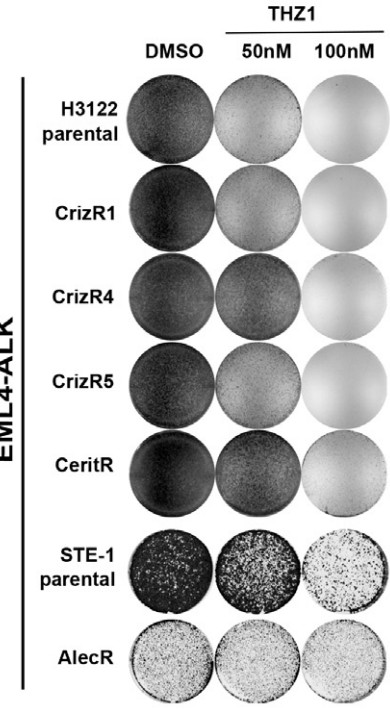

**C**

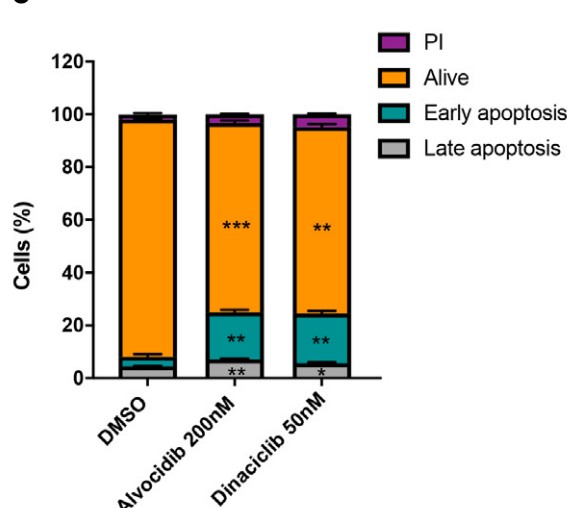

Figure 3.

**Figure 3. Cells harbouring the EML4/ALK translocation are remarkably sensitive to alvocidib, dinaciclib and THZ1.**

A  Proliferation assay of CrizR1, CrizR4, CrizR5, CeritR and AlecR isogenic drug-resistant cell lines, treated with the indicated concentrations of THZ1 for 72 h. $P < 0.05$ was calculated for $IC_{50}$ shift, as indicated in the Materials and Methods.
B  Crystal violet staining of EML4/ALK parental and drug-resistant cells. Cells were treated with the indicated drugs until the vehicle control reached confluence, then fixed and stained.
C  CrizR1 cells were treated with the indicated drugs for 72 h, and then, cells were stained with Annexin V/PI. Flow cytometry was then used to quantify Annexin V+ cells. Alvocidib: early apoptosis $P = 0.002$, late apoptosis $P = 0.006$, alive $P = 0.0008$; Dinaciclib: early apoptosis $P = 0.005$, late apoptosis $P = 0.02$, alive $P = 0.002$.

Data information: Statistical comparisons were performed using a paired, two-tailed Student $t$-test. Plotted graphs show mean $\pm$ SD ($n = 3$). *$P < 0.05$, **$P < 0.01$, ***$P < 0.001$, N.S. = Not Significant $P > 0.05$.

(Figs 4D and EV3E). Conversely, CDK7 knockdown on its own did not lead to apoptosis, suggesting a need for combined inhibition of CDK7/12/13.

Interestingly, levels of some of the previously examined cell cycle-related genes were upregulated in CrizR1 compared with the H3122 parental cells and were significantly reduced upon treatment with alvocidib or the transcription inhibitor actinomycin D, corroborating the hypothesis of a transcriptional regulation (Fig 4E).

Subsequently, we hypothesized that the short-lived, anti-apoptotic mRNAs such as *MCL-1* and *Survivin* are degraded after transcriptional inhibition. Consistently, *MCL-1* and *Survivin* were downregulated at the mRNA level after treatment with all the compounds (Fig 4F). We asked whether *MCL-1* or *Survivin* downregulation was enough to induce apoptosis and to account for alvocidib-induced cell death. We silenced *MCL-1* or *Survivin* using two different siRNAs for *MCL-1* and a pool of 4 different siRNAs for *Survivin*, and we observed a significant induction of apoptosis upon *Survivin* and not *MCL-1* silencing, suggesting that *Survivin* downregulation is partly responsible for the apoptotic response to CDK inhibitors. (Fig 4G and Appendix Fig S3A). To shed light on the specificity of these compounds towards EML4-ALK cells, we analysed RNA-seq expression data from the cancer cell line encyclopaedia (CCLE) (Ghandi *et al*, 2019). We used the HALLMARK apoptosis gene collection and plotted the z-score of the apoptotic genes in all the lung adenocarcinoma lines (LUAD) (Appendix Fig S3B). Intriguingly, H3122 cells had the highest expression levels of *Cyclin D1* as well as *MCL-1* compared with the rest of the LUAD cells (Fig 4H and Appendix Fig S3C). *Survivin* is not part of the HALLMARK gene set, but when we looked at it separately, we found that H3122 cells had the highest *Survivin* mRNA levels (Fig 4H). Altogether, these findings indicate that treatment with alvocidib or THZ1 leads to cell death at least in part through *Survivin* downregulation.

**Effects of alvocidib and THZ1 treatment on transcription initiation and elongation**

In order to add confidence to the transcriptional hypothesis, we performed ChIP-seq for RNA polymerase II after treating CrizR1 cells with alvocidib or THZ1. A global overview of RNA pol II peaks suggested that alvocidib treatment dramatically increased occupancy at the transcription start site (TSS), while THZ1 decreased it (Fig 5A). We performed GSEA analysis based on the core enrichment of the mapped peaks and found 6 differentially enriched signatures with alvocidib (Fig 5B) and 11 with THZ1 (Fig 5C). Notably, both drugs induced different RNA pol II occupancy in the TSS of MYC targets. Lastly, we looked at the peaks of the previously examined genes *MCL1, Survivin* and *CCND1* and we

also included *MYC*. From the gene tracks, it was evident that alvocidib treatment resulted in pausing of the RNA pol II at the TSS while reducing the occupancy across the gene bodies, while THZ1 resulted in reduced binding of RNA pol II at the TSS and in the gene bodies (Fig 5D). This is highly concordant with previous findings which suggest that CDK7 mediates the binding of the RNA pol II at the promoters while CDK9 regulates the release and elongation steps (Kwiatkowski *et al*, 2014). *MYC* and *CCND1* downregulation upon alvocidib or THZ1 treatment was confirmed by qPCR (Fig 5E). Notably, *MYC* silencing induced upregulation of the pro-apoptotic proteins BIM and significantly increased cell death (Fig 5F and G). Furthermore, MYC indirect inhibition via trametinib, rapamycin or both significantly induced cell death in CrizR1 cells (Fig 5H). In conclusion, transcriptional inhibition with the described CDK inhibitors offers a new way to induce apoptosis in EML4-ALK lung cancer cells.

**Alvocidib is effective *in vivo***

To test the activity of alvocidib *in vivo*, we first characterized the resistant cells to verify whether they could keep the resistance in the absence of the drugs for up to 6 weeks. CrizR1, CrizR4 and AlecR cells showed marked resistance in the absence of crizotinib for up to 6 weeks (Fig EV4A), although the AlecR cells are primarily more sensitive to the drug compared with the crizotinib-resistant cells. Next, we tested the activity of alvocidib *in vivo*. As expected, H3122 xenograft mouse models responded to crizotinib and alvocidib (Fig EV4B and C). CrizR1 and CrizR4 xenograft mouse models were resistant to crizotinib while treatment with alvocidib resulted in reduced tumour growth (Figs 6A and B, and EV4D and F). We also noticed a significant decrease in tumour growth in response to alvocidib in mice harbouring alectinib-resistant tumours (Figs 6C and EV4G). Alvocidib treatment was generally well tolerated, although weight loss was observed in some of the mice (Appendix Fig S4). Upon sacrificing the mice, we assessed the apoptotic status of these tumours by *in situ* immunohistochemistry (IHC). The increase in cleaved caspase 3 was evident in CrizR1 and AlecR tumours (Fig 6D), while in CrizR4 tumours the levels of cleaved caspase 3 were marginally significant (Fig EV5A). However, we should note that in the CrizR4 model the absence of drug treatment during the last week before sacrifice complicates this interpretation.

In agreement with *in vitro* studies, in tumours treated with alvocidib we observed downregulation of the marker of proliferation Ki67 as well of *MYC, Survivin* and *MCL-1*, and upregulation of *BIM* and p21(Figs 6E and F, and EV5B and C).

These results raised questions in terms of sequential use of ALK inhibitors. For patients with wild-type ALK who progress on

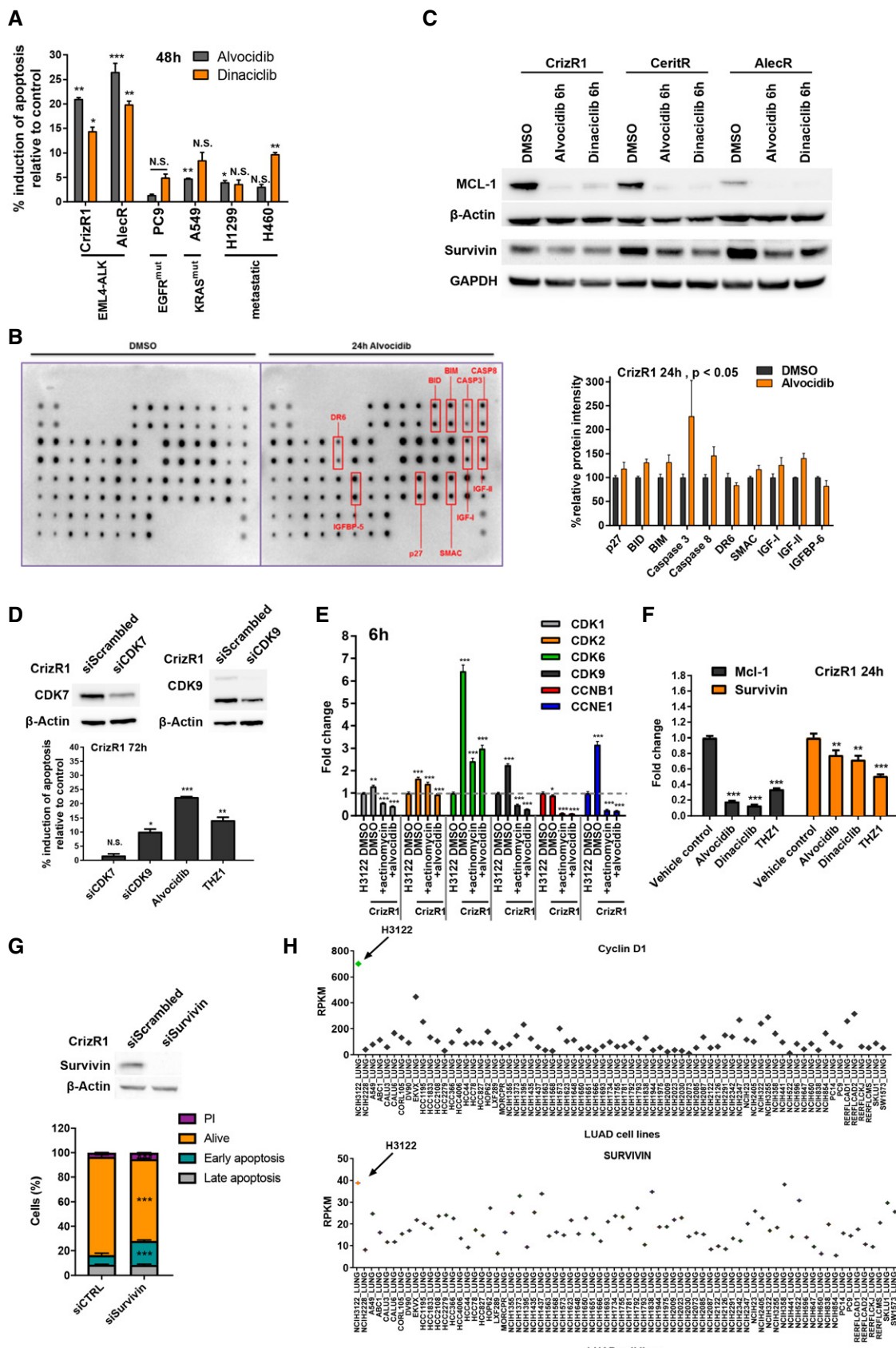

**Figure 4.**

◀

**Figure 4. Alvocidib induces cell death through the mitochondrial apoptotic pathway.**

A   Indicated cell lines were treated with DMSO, 200 nM alvocidib or 25 nM dinaciclib for 48 h, then cells were stained with Annexin V/PI. Flow cytometry was then used to quantify Annexin V+ cells (n = 2). Alvocidib: CrizR1 P = 0.005, AlecR P = 0.001, PC9 P = 0.05, A549 P = 0.002, H1299 P = 0.03, H460 P = 0.07; Dinaciclib: CrizR1 P = 0.02, AlecR P = 0.005, PC9 P = 0.06, A549 P = 0.08, H1299 P = 0.09, H460 P = 0.01.

B   CrizR1 cells were treated with DMSO or 200 nM alvocidib for 24 h, and cell extracts were hybridized to a 43-antibody array and analysed by immunoblotting. Graphs depict the significant changes from two independent experiments.

C   CrizR1/CeritR/AlecR cells were treated with DMSO or 200 nM alvocidib or 50 nM dinaciclib for 6 h and cell extracts were analysed by Western blotting.

D   (Top) Western blot analysis of CrizR1 cells treated with siScrambled or siRNA for CDK7 or CDK9 for 72 h. (Bottom) CrizR1 were treated as above, or with DMSO/alvocidib/THZ1 and cells were stained with Annexin V/PI and analysed by flow cytometry for Annexin V+ cells 72 h post-transfection (n = 2). Annexin siCDK7 P = 0.2, siCDK9 P = 0.015, alvocidib P = 0.0001, THZ1 P = 0.004.

E   H3122 parental and CrizR1 cells were treated with DMSO, 200 nM alvocidib or 250 ng/ml actinomycin D for 6 h. RNA was extracted, and the mRNA levels of the indicated genes were quantified by RT-qPCR. CDK1: DMSO versus H3122 DMSO P = 0.002, Actinomycin versus DMSO P = 0.0003, Alvocidib versus DMSO P = 0.0005; CDK2: DMSO versus H3122 DMSO P = 0.0001, Actinomycin versus DMSO P < 0.0001, Alvocidib versus DMSO P = 0.0005; CDK6: DMSO versus H3122 DMSO P = 0.0005, Actinomycin versus DMSO P = 0.0004, Alvocidib versus DMSO P = 0.0004; CDK9: DMSO versus H3122 DMSO P = 0.0003, Actinomycin versus DMSO P = 0.0002, Alvocidib versus DMSO P = 0.0003; CCNB1: DMSO versus H3122 DMSO P = 0.02, Actinomycin versus DMSO P = 0.0002, Alvocidib versus DMSO P = 0.0003; CCNE1: DMSO versus H3122 DMSO P = 0.0004, Actinomycin versus DMSO P = 0.0004, Alvocidib versus DMSO P = 0.0005.

F   RT-qPCR analysis of *MCL-1* and *Survivin* expression after treatment of CrizR1 cells with 100 nM alvocidib, 25 nM dinaciclib and 50 nM THZ1. RNA was extracted after 24 h of treatment. MCL-1 alvocidib versus control P = 0.0001, dinaciclib versus control P = 0.0001, THZ1 versus control P = 0.0001; Survivin: alvocidib versus control P = 0.01, dinaciclib versus control P = 0.01, THZ1 versus control P = 0.0001.

G   (Top) Western blot analysis of CrizR1 cells treated with siScrambled or with a pool of 4 different siRNAs targeting *Survivin* (siBIRC5). (Bottom) Cells were stained with Annexin V/PI and analysed by flow cytometry for Annexin V+ cells 72 h post-transfection. Annexin: early apoptosis P = 0.0003, late apoptosis P = 0.5, alive P < 0.0001.

H   The CCLE RNA-seq data set was used and RPKM values were plotted for the *Cyclin D1* and *Survivin* genes, indicating high expression in H3122 cells compared with other LUAD cells.

Data information: Statistical comparisons were performed using a paired, two-tailed Student t-test. Plotted graphs show mean ± SD (n = 3, unless otherwise specified).
*P < 0.05, **P < 0.01, ***P < 0.001, N.S. = Not Significant P > 0.05.
Source data are available online for this figure.

crizotinib or alectinib, the third-generation ALK inhibitor lorlatinib could conceivably be used to more potently inhibit ALK. In a proliferation assay, while H3122 parental cells were very sensitive to lorlatinib-, crizotinib- and ceritinib-resistant cells were cross-resistant to lorlatinib while they were very sensitive to alvocidib or dinaciclib (Fig 6G). Alectinib-resistant cells did respond to lorlatinib, but they were more sensitive to transcriptional inhibition. However, resistance of CrizR1 cells to lorlatinib was not confirmed *in vivo* in a mouse xenograft model (Figs 6H and EV5D), possibly due to the high concentration of lorlatinib used *in vivo* compared with the concentration used *in vitro*. Therefore, we propose to use lorlatinib as second-line therapy in patients with wild-type ALK who became refractory to crizotinib, and alvocidib when resistance to ALKi occurs as potential alternative to chemotherapy. In conclusion, we have presented a potential new alternative to chemotherapy in the refractory setting (Fig 6I).

## Discussion

In this paper, we have found that a global transcriptional dysregulation leads to crizotinib resistance in EML4-ALK cells. Furthermore, we have shown that transcriptional inhibition is highly potent in this context and that the CDK inhibitors alvocidib, dinaciclib and THZ1 warrant clinical testing in patients with disease progression due to ALK-independent mechanisms of resistance. Specifically, we have shown that downregulation of key cell cycle proteins as well as an upregulation of pro-apoptotic proteins mediate the response to crizotinib.

In our data, even though EGFR overexpression and activation was present in CrizR1 cells, it did not mediate resistance to crizotinib. This is not surprising, given that a case of crizotinib-resistant cells with EGFR amplification that had another main driver of

resistance has been reported before (Katayama *et al*, 2012). TGF-β inhibition had a marginal effect on the proliferation of crizotinib-resistant cells, implying that TGF-β activity acts in part to promote resistance to crizotinib. Furthermore, although AXL inhibition partially reduced cell proliferation it had no effect in inducing programmed cell death in ALK+-resistant cells.

Amplification of CDKs or their partner Cyclins has been described to confer a proliferative advantage to tumour cells (Otto & Sicinski, 2017). Recent work in EGFR-mutant lung cancer revealed that amplification of the *CCNE1* gene can be found in patients with acquired resistance to EGFR inhibitors (Blakely *et al*, 2017) and specifically with acquired resistance to osimertinib (Yang *et al*, 2018). We have demonstrated that cells with acquired resistance to ALK inhibitors present CDKs/Cyclins upregulation. We thereby observed the impressive single-agent activity of CDK inhibitors that cannot be explained by single inhibition of CDK1, CDK2 or CDK6, as evidenced by siRNA experiments. This previously unseen potent activity of CDK inhibitors at concentrations that are easily achievable in the clinic (Shapiro *et al*, 2001; Stephenson *et al*, 2014) prompted us to investigate this further.

Alvocidib and dinaciclib are known to potently inhibit the transcriptional regulator CDK9 (Parry *et al*, 2010), while THZ1 inhibits the transcriptional regulators CDK7/12 (Kwiatkowski *et al*, 2014). CDK7/12 and CDK9 regulate transcription by phosphorylating the RNA polymerase II (Oelgeschläger, 2002). With siRNA experiments, we were able to induce apoptosis by partial CDK9 downregulation. CDK7 knockdown did not lead to apoptosis, but it has been observed before that concurrent inhibition of CDK7/12/13 is required for an effect on transcription (Olson *et al*, 2019).

While phosphorylation of RNA pol II at the Ser5 and Ser7 sites by CDK7 has been shown to be important for the recruitment of the complex at the TSS (Sampathi *et al*, 2019), phosphorylation at Ser2 by CDK9 is important for the release and the elongation step

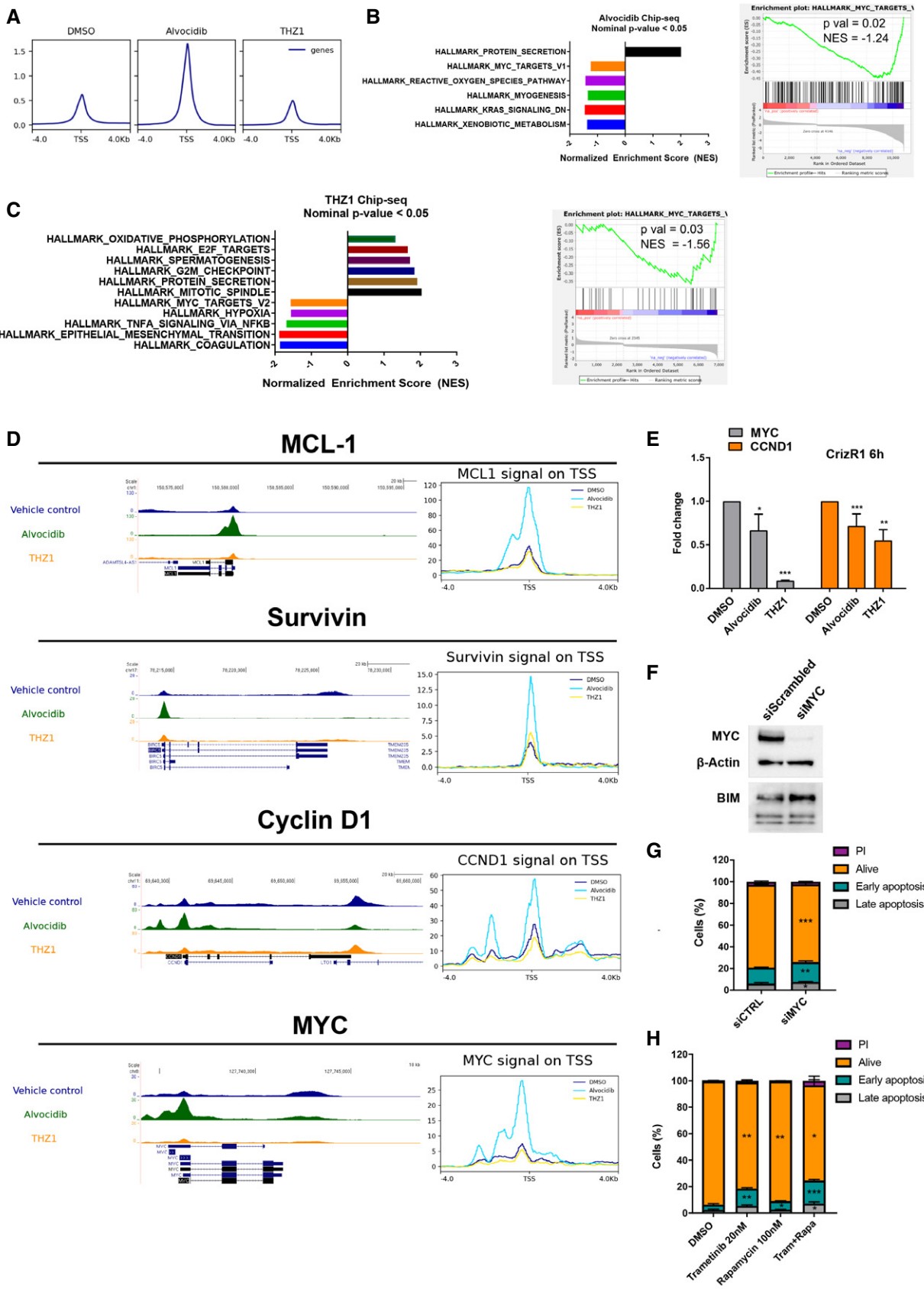

Figure 5.

**Figure 5. Alvocidib or THZ1 treatment is concordant with CDK9 or CDK7 inhibition based on RNA polymerase II ChIP-seq.**

A   CrizR1 cells were treated with vehicle control, 200 nM alvocidib or 100 nM THZ1 for 6 h, then chromatin was precipitated with an anti-RNA polymerase II antibody and sequenced. Plotted is the average number of peaks per condition.

B, C   GSEA analysis using the HALLMARK gene collection for the differentially enriched peaks around the TSS ($\pm$1 kb) with (B) alvocidib or (C) THZ1 treatment. Shown on the right is the plot of the "MYC targets"-enriched signature.

D   Gene tracks of RNA polymerase II occupancy at the *MCL1*, *BIRC5* (*Survivin*), *CCND1* and *MYC* genes.

E   qPCR of *MYC* and *CCND1* upon treatment of CrizR1 cells with Alvocidib or THZ1 for 6 h. The graph represents the mean fold change $\pm$ SD. MYC: alvocidib versus DMSO $P = 0.03$, THZ1 versus DMSO $P < 0.0001$; CCND1: alvocidib versus DMSO $P = 0.001$, THZ1 versus DMSO $P = 0.003$.

F   BIM is upregulated upon MYC silencing in CrizR1 cells.

G   Annexin V assay showing the percentage of apoptotic CrizR1 cells upon MYC downregulation using siRNA. Early apoptosis $P = 0.002$, late apoptosis $P = 0.04$, alive $P = 0.0008$.

H   Annexin V assay showing the percentage of apoptotic CrizR1 cells after treatment with trametinib, rapamycin or both. Trametinib: early apoptosis $P = 0.002$, late apoptosis $P = 0.05$, alive $P = 0.009$; rapamycin: early apoptosis $P = 0.04$, late apoptosis $P = 0.8$, alive $P = 0.003$; tram+rapa: early apoptosis $P = 0.0001$, late apoptosis $P = 0.04$, alive $P = 0.01$.

Data information: Statistical comparisons were performed using a paired, two-tailed Student *t*-test. Plotted graphs show mean $\pm$ SD ($n = 3$, unless otherwise specified). *$P < 0.05$, **$P < 0.01$, ***$P < 0.001$.

Source data are available online for this figure.

(Jonkers *et al*, 2014). Through ChIP-seq, we demonstrated that alvocidib treatment results in RNA Pol II promoter-proximal pausing, while THZ1 treatment decreases the occupancy of PolII at TSS. These results suggest that in the described cellular context, inhibition of CDK7 and CDK9 has a widespread transcriptional effect leading to downregulation of the transcription factor MYC and its family members along with several other oncogenes, including *CCND1* and S*urvivin*.

Our proposed model of action for these compounds is an induction of apoptosis independent of cell cycle arrest since in our data it is clear that cell cycle arrest is minimal compared with apoptotic induction. We posit that there is a p53-independent induction of apoptosis since the parental H3122 cells harbour the E285V inactivating mutation of the *TP53* gene (COSMIC project, Sanger Institute and (Russell-Swetek *et al*, 2008)). Furthermore, we suggest that the downregulation of RNA polymerase II activity promotes the loss of short-lived mRNAs, such as *MYC* and *Survivin* (Blagosklonny, 2004) and an upregulation of pro-apoptotic genes followed by initiation of apoptosis and specifically of the mitochondrial apoptotic pathway. Notably, the increase of the pro-apoptotic gene BIM in this context appears to be, at least in part, MYC-dependent.

We confirmed that alvocidib and dinaciclib induced significantly higher levels of apoptosis in crizotinib-resistant cells compared with KRAS$^{mut}$ and metastatic NSCLC cell lines or non-transformed epithelial cells. This could explain why, in terms of clinical testing, alvocidib was not effective in a previous clinical trial on advanced metastatic NSCLC (Shapiro *et al*, 2001). Dinaciclib was shown to be inactive as monotherapy in patients with NSCLC previously treated with erlotinib (Stephenson *et al*, 2014). However, the ALK mutational status in these patients was not assessed; therefore, the trial did not test the patient cohort that our data would represent, namely patients with EML4-ALK NSCLC. Consistent with the compelling *in vitro* activity, alvocidib was also active in xenograft models of crizotinib and alectinib resistance, where ALK inhibitors failed. To our knowledge, this is the first time this has been demonstrated in a xenograft mouse model of lung adenocarcinoma.

In conclusion, we have reinforced the idea that the appearance of a known oncogene as mutated or over-activated does not necessarily mean that this oncogene is the main driver of drug resistance. We have shown multiple alterations that concurrently occur in the resistance to ALK inhibitors. This complicates the diagnostic setting and suggests that testing for individual ALK-independent mechanisms of resistance in the clinic could be inefficient. We can therefore envision that a more efficient way to address off-target TKI resistance is either with rational upfront combinations that aim to prevent it altogether (Hrustanovic *et al*, 2015; Rusan *et al*, 2018), or using a drug with universal activity able to dampen parallel oncogenic pathways simultaneously.

**Figure 6. Alvocidib reduces tumour growth *in vivo*.**

A, B   Tumour growth of *in vivo* xenografts of H3122 CrizR1 cell lines in response to crizotinib or alvocidib and correspondent tumour weights (P.O. control $n = 5$; I.P. control $n = 4$; crizotinib $n = 8$ and alvocidib $n = 6$). Tumour weights, crizotinib $P = 0.44$, alvocidib $P = 0.03$.

C   Tumour growth of *in vivo* xenografts of Ste-1 AlectR cell lines in response to alvocidib and correspondent tumour weights (I.P. control $n = 6$; alvocidib $n = 4$). Tumour weights alvocidib $P = 0.004$.

D   Number of cleaved caspase-3-positive nuclei in alvocidib treated tumours compared with controls.

E   Ki67 and p21 staining of CrizR1 xenograft tumours. Scale bar = 200 μm. Quantitative analysis of IHC staining is presented on the right. Ki67 $P = 0.0004$; p21 $P = 0.02$.

F   qPCR of *Survivin* ($P = 0.0009$), *MCL-1* ($P = 0.04$), *MYC* ($P = 0.007$) and *BIM* ($P = 0.03$) in Ste-1 AlectR cells xenografts treated with alvocidib (Control $n = 6$; alvocidib $n = 4$).

G   Proliferation assay of H3122 parental and isogenic drug-resistant cell lines, treated with DMSO, 100 nM lorlatinib, 200 nM alvocidib or 50 nM dinaciclib for 72 h ($n = 4$).

H   Growth curve of CrizR1 xenograft tumours in response to lorlatinib (Control $n = 7$; lorlatinib $n = 7$).

I   Model suggesting the clinical sequencing of CDK inhibitors post-ALK inhibition failure as a potential alternative to chemotherapy.

Data information: Statistical comparisons were performed using a paired, two-tailed Student *t*-test. Plotted graphs show mean $\pm$ SD. *$P < 0.05$, **$P < 0.01$, ***$P < 0.001$, N.S. = Not Significant $P > 0.05$.

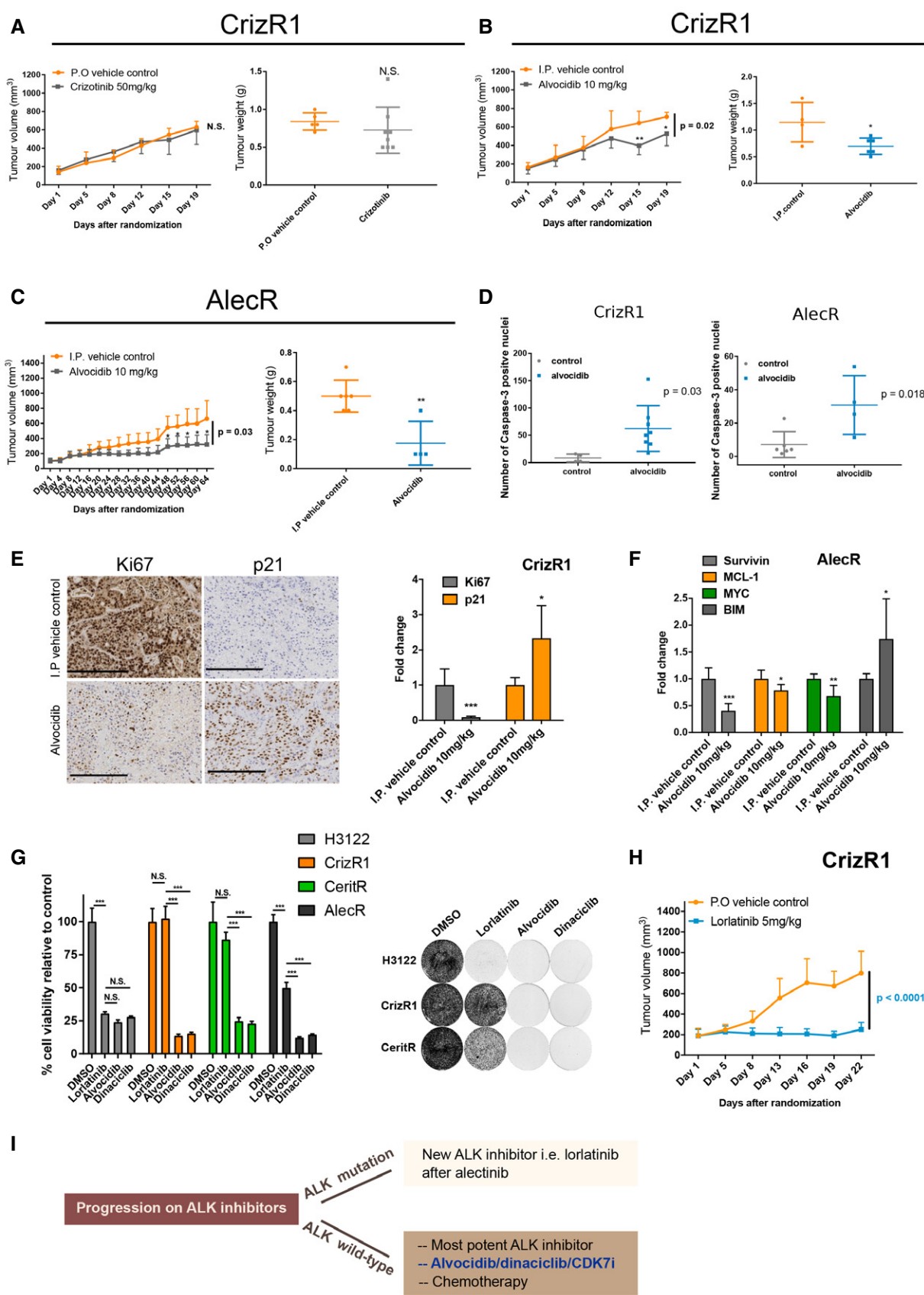

Figure 6.

# Materials and Methods

## Apoptotic assay

### Annexin V/PI

Cells were trypsinized and processed with a TACS® Annexin V/PI kit (Trevigen) according to the manufacturer's instructions. Samples were then run through a NovoCyte flow cytometer (ACEA biosciences) and quantified for Annexin V+ cells. Spectral overlap was compensated with unstained and single stained controls. Plots were gated to exclude cell debris and cell doublets.

## Senescence assay

Cellular senescence was detected measuring the activity of the β-Galactosidase (β-Gal) using CellEvent Senescence Green Detection kit (Invitrogen). 2,000 cells/well were plated in Nunc 96-well optical plates. At 72 h after treatments, cells were washed with PBS, fixed by adding 100 μl/well of fixation solution for 10 min and then washed again with 100 μl of 1% BSA in PBS. Next, 100 μl/well of the pre-warmed working solution, containing the fluorescence-based β-gal substrate, were added to each well and incubated for 2 h at 37°C. Cells were washed 3 times with PBS and, finally, fluorescence by β-gal-cleaved substrate was measured at 488 nm with M5 SpectraMax plate reader.

## Cell culture

HEK293T cells were maintained in DMEM with 1 g/l glucose (Gibco) + 10% fetal bovine serum (Gibco). HBEC cells were maintained in Airway Epithelial Cell Basal Medium (ATCC) combined with a Bronchial Epithelial Cell Growth kit (ATCC). All the other cell lines were maintained in RPMI-1640 (Gibco) + 10% fetal bovine serum (Gibco).

HEK293T and A549 cells were a kind gift from Dr. John Brognard. H1299 cells were a kind gift from Prof. Angeliki Malliri. HBEC, H2228 and H460 cells were purchased from ATCC. The following cell sources have been described before: H3122 (Lovly *et al*, 2011), STE-1 (Lovly *et al*, 2014), PC-9 and isogenic cell lines (Meador *et al*, 2015), H1975 and isogenic cell lines (Meador *et al*, 2015). H3122 and STE-1 parental cells were STR-profiled upon receipt and found to be free of contamination with another line. All cell lines were routinely monitored for mycoplasma contamination using an in-house Core Facility service.

## Drug-resistant cell lines

Full information for drug-resistant cells can be found in Fig 1A. Briefly, the parental cell lines were exposed to a low concentration of the primary inhibitor until cells could proliferate fully in the presence of it. Then cells were split and exposed to 20% higher concentration. The process was repeated until the corresponding cell lines were resistant to > 500 nM of the primary inhibitor and were then maintained by adding drug after every passage.

## Characterization of resistant cell lines

The capacity of the cells to maintain resistance to crizotinib and alectinib was assessed over a period of 6 weeks. Briefly, resistant cell lines (CrizR1, CrizR4 and AlecR) were split to generate a subline of cells cultured in the absence of drug (−drug), in parallel to the original resistant cell line cultured in the presence of the drug (+drug). Cell viability was assessed every week until 6 weeks upon treatment with crizotinib (1 μM) or alectinib (500 nM) for 72 h.

## Cell cycle analysis

After the corresponding treatment, cells were fixed in 70% ethanol and stained with an FxCycle™ PI/RNase Staining Solution (Molecular Probes). Cell cycle plots were generated in a Novocyte instrument using the device's software.

## Compounds used

Crizotinib, ceritinib, alectinib, erlotinib, osimertinib, palbociclib, alvociclib, dinaciclib, THZ1, galunisertib and trametinib for *in vitro* experiments were all purchased from Selleck. Lorlatinib was purchased from MedChemExpress. Bemcentinib (R428) was purchased from Axon. Rapamycin was purchased from Caymen Chemical. Actinomycin D was purchased from Sigma. Alvociclib, dinaciclib, crizotinib, lorlatinib and THZ1 for *in vivo* studies were purchased from MedChemEexpress whereas alectinib was purchased from Selleck.

## Computational analyses

### GSEA

Gene set enrichment analysis was performed by ranking the gene sets after differential expression analysis of RNA-seq data. The cut-off for the final list of genes was $P$ adj < 0.05 and log2foldchange > 1 or < −1. The.rnk files were loaded to the supplied desktop GSEA v2.0 software (Subramanian *et al*, 2005). The analysis parameters were based on 1,000 permutations and a minimum of 15 genes per identified set. The individual gene set collections used are described in the text or figure legends.

### Heatmap

Heatmap was generated using the R Package pHeatmap based on Euclidean distance clustering.

## Crystal violet staining

Cells were treated in 6-well plates, fixed for 20 min in 3.7% paraformaldehyde and then stained for 20 min in a 0.05% crystal violet solution (Sigma).

## In vivo xenograft studies

Female athymic nude mice or NOD-SCID mice from Charles River were used at 6–8 weeks old. After acclimatization, mice were injected with 2 million H3122 cells in 50% matrigel, or 2 million CrizR1 cells in 50% matrigel, 2.5 million CrizR4 cells or 5 million AlecR cells in 50% matrigel (Corning). When tumours reached between 100 and 200 mm$^3$, mice were randomized in different groups.

H3122/CrizR1 experiment: 10 ml/kg vehicle control by oral gavage daily ($n$ = 5), 50 mg/kg crizotinib by oral gavage daily

($n = 8$), 10 ml/kg vehicle control by I.P. injection 3× a week ($n = 5$), 10 mg/kg alvocidib by I.P. injection 3× a week ($n = 8$). In the H3122 experiment, the tumour size of the control group and the gavage group were combined in the graph to facilitate comparison with crizotinib. For the CrizR1 experiment, measurements shown are 3 days before sacrifice.

CrizR4 experiment: Initially, mice were randomized to receive 10 ml/kg vehicle control ($n = 8$) or 10 mg/kg alvocidib ($n = 8$), by I.P. injection, daily for 3 weeks. In the crizotinib group, mice were treated with 10 ml/kg vehicle control or 50 mg/kg crizotinib daily, by oral gavage.

AlecR experiment: Initially, mice were randomized to receive 10 ml/kg vehicle control by I.P. injection 3×/week ($n = 7$), 10 mg/kg alvocidib by I.P. injection 3×/week ($n = 7$). After 3 weeks of consecutive treatment, a 1 week on/1 week off pattern was adopted for the alvocidib/I.P. control group.

CrizR1 experiment with lorlatinib treatment: 10 ml/kg vehicle control by oral gavage daily ($n = 8$), 5 mg/kg lorlatinib by oral gavage daily ($n = 8$).

Tumours were measured with a digital calliper, and volumes were calculated using the formula [volume = $(width)^2$ $(length)/2$. In all the experiments, mice that showed weight loss > 20% of body weight or with tumours that reached a size of 1,500 $mm^3$ or that developed tumour ulceration/bleeding were sacrificed and discounted from the graphs from that point onwards. In all experiments, mice in the control group that did not develop tumours bigger than 300 $mm^3$ by the end of the study were discounted. Crizotinib and alvocidib were dissolved in 5% DMSO, 40% PEG300 and 55% sterile PBS, and the same mix was used as vehicle control. All procedures involving animals were approved by the CRUK Manchester Institute's Animal Welfare and Ethical Review Body.

### IHC

Tumours in formalin were embedded in paraffin blocks, and different slices were obtained. Slides were incubated with an anti-cleaved caspase-3 antibody (Cell Signaling (#9661) at 1/100 dilution; anti-p21 (Abcam #109520) at 1/100 dilution and anti-Ki67 (Abcam #15580). This was run on a Leica bond Rx with the Refine Kit with an added casein blocking step with the antigen retrieval ER1 at pH 6 for 20 min.

#### Quantification

One section per tumour was imaged at 20×, and the number of stain-positive cells was quantified in 5 random fields (One field contains approximately 1,600 cells). The mean number of positive cells was then plotted.

### RNA isolation

RNA was isolated using TRIzol® Reagent (Ambion), according to the manufacturer's protocol.

### RNA profiling

#### RNA-sequencing

*Poly-A*: Poly-A libraries were prepared with a SureSelect PolyA kit (Agilent). RNA-seq reads were quality checked and aligned to the

human genome assembly (GRCh37/hg19). Differential expression (DE) was evaluated using the DESeq2 package.

### ChIP-seq

CrizR1 cells ($100 \times 10^6$ per condition) were treated with DMSO, 200 nM alvocidib or 100 nM THZ1 for 6 h. Then, chromatin was isolated according to the published protocol (Nelson *et al*, 2006) using 10 μg of total RNA polymerase II antibody (Diagenode, #C15200004). Then, purified chromatin was used as input to generate PCR-amplified libraries using the Diagenode Microplex kit according to the manufacturer's instructions. The library was sequenced on a Nextseq500 instrument (Illumina) using paired-end sequencing at a sequencing depth of 60–80 million reads per sample. Afterwards, the quality of the sequenced reads was assessed by the FastQC (Babraham Institute 2010)/fastq-screen (Babraham Institute 2011) output supplied by the CRUK computational biology facility. Reads were filtered using Trimmomatic v0.36 (Bolger *et al*, 2014), to remove any remaining adapter sequences, poor quality 5′ ends of reads or reads shorter than 35 nucleotides. Reads were then mapped to the human genome (UCSC GRCh38/hg38 analysis set) using Bowtie2 v2.3.0 (Langmead & Salzberg, 2012). Mapped reads were filtered to retain concordant read pairs with a mapping quality of at least 30, using samtools v1.9 (Li *et al*, 2009 The Sequence Alignment/Map format and SAMtools). The peaks were called by the use of MACS2 v2.1.2 (Zhang *et al*, 2008) with the default parameters. Only binding regions with a Qvalue of < 0.05 were considered. Differential binding analysis was performed using diffReps v1.55.6 (Shen *et al*, 2013) using the midpoint coordinate of the filtered mapped paired-reads, with fragment size set to 0. RnaChipIntegrator https://github.com/fls-bioinformatics-core/RNAChipintegrator was used to identify the closest gene within 100 K of each peak; the distance being calculated between the closest edge of the summit region and the TSS of each gene.

### RT-qPCR

Total RNA was extracted using TRIzol. For gene-specific qPCR, 200 ng of total RNA were used as input with the Verso cDNA synthesis kit (Thermo Fisher Scientific) according to the manufacturer's protocol. Then, 1 μl cDNA from each reaction was amplified with the FS Universal SYBR Green Master Rox master mix (Roche). The reaction was run on a LightCycler 96 instrument (Roche) and normalized for relative expression using either ACTB (β-actin)or GAPDH as housekeeping genes. All the primers used were custom-designed and can be found in Table EV1.

### Proliferation assay

3,000–5,000 cells were plated in 96-well plates and cell proliferation was assessed by adding 20 μl/well of the CellTiter 96® Aqueous Non-Radioactive Cell Proliferation reagent (Promega). Absorbance was recorded at 490 nm using a SpectraMax plate reader. Absorbance was normalized to the control and was fitted using a non-linear regression curve (Prism 7, GraphPad).

### Statistical analysis

All statistics and graphs were generated using Prism 7 (GraphPad). Plotted graphs show mean ± SD from 3 biological replicates for all

experiments unless indicated otherwise in the figure legend. Statistical comparisons were performed using a paired, two-tailed Student $t$-test where $*P < 0.05$, $**P < 0.01$ and $***P < 0.001$, N.S. = Not Significant. For linear regression analyses (indicated in the legends), values were log-transformed and normalized, and then curves were compared for $IC_{50}$ difference with an extra-sum-of-squares F-test.

For RNA-seq analysis, the resulting $P$-values were adjusted using the Benjamini and Hochberg approach. Genes with an adjusted $P$-value determined to be $< 0.05$ (FDR $< 0.05$) by DESeq2 and that had a fold change value $\geq 1.5$ (|Log2 fold change|$\geq 0.55$) between two groups were considered to be differentially expressed.

### Transfection

All transfection experiments were performed using Lipofectamine 2000 or Lipofectamine RNAiMax (Invitrogen). Smartpool siRNAs for EGFR, CDK1, CDK2, CDK6, CDK7 and CDK9 or siRNA controls were obtained from Dharmacon and transfected at 33–100 nM final concentration. siRNAs for MCL-1 were obtained from Qiagen and transfected at 25 nM final concentration. A pool of 4 different siRNAs (SMARTpool) for Survivin were purchased from Dharmacon and transfected at 100 nM final concentration.

### Western blotting

Protein extracts were isolated using a RIPA buffer (Sigma Aldrich) and quantified with the use of the BCA-pierce assay (Thermo Fisher). 30–50 μg of proteins were combined with a Novex™ Tris-Glycine SDS Sample Buffer (2×) (Invitrogen) and NuPAGE™ Sample Reducing Agent (10×) (Invitrogen). Samples were then heated at 70°C for 10 min. Afterwards, samples were loaded to either 4–12% or 3–8% NuPage gels (Invitrogen) and run with NuPAGE™ MOPS SDS or NuPAGE™ Tris-Acetate SDS Running Buffer (Invitrogen). Proteins were then transferred to a PVDF membrane (Immobilon) and blocked in 5% non-fat dry milk. Membranes were then incubated O/N at 4°C with the indicated antibodies. Membranes were then incubated with secondary HRP-conjugated antibodies (Amersham) and developed using a WesternBright ECL Spray (Advansta). Chemiluminescence was recorded in a Bio-Rad Chemidoc instrument. The antibodies used can be found in Table EV1. For the apoptotic array, the assay was performed according to the manufacturer's instructions (Abcam).

## Data availability

The data sets produced in this study are available in the following databases: RNA-seq data: Array Express repository E-MTAB-8590 (https://www.ebi.ac.uk/arrayexpress/experiments/E-MTAB-8590); Chip-Seq data: Array Express repository E-MTAB-8380 (https://www.ebi.ac.uk/arrayexpress/experiments/E-MTAB-8380).

Expanded View for this article is available online.

### Acknowledgements
We are grateful to the Imaging & Cytometry, Histology, Computational Biology and Molecular Biology core facilities in the CRUK Manchester Institute for their help. Work in the MG laboratory is funded by a CRUK MI core grant (C5759/A20971).

### The paper explained

#### Problem
Patients with EML4-ALK–driven lung cancer are treated with targeted therapies. While initial responses are excellent, patients eventually relapse due to the development of acquired resistance to these therapies.

#### Results
Drug-resistant EML4-ALK cells are sensitive to treatment with certain CDK inhibitors. This treatment induces apoptosis and reduces tumour growth in mouse models of the disease.

#### Impact
Patients with acquired resistance to ALK inhibitors and wild-type ALK have only chemotherapy left as treatment option. These findings suggest that CDK inhibition may be clinically tested as an alternative to help manage this disease.

## Author contributions

ARP, MB designed and performed experiments, analysed the data, generated the figures and wrote the manuscript. LS, performed *in vivo* and *in vitro* experiments. PM and GDL performed *in vitro* experiments. SS, IJD and H-SL were responsible for bioinformatics analyses. MF performed the *in vivo* tumour staining score. FB, MK and MC designed part of the study and provided feedback. CML was responsible for the generation and characterization of the acquired resistance cell line models. MG conceived the study, supervised experiments and wrote and edited the manuscript.

## Conflict of interest

None of the conflicts of interest pertain to this manuscript. Otherwise, F. Blackhall declares research funding and honoraria from MSD, Novartis, Amgen, AbbVie, AstraZeneca, Pfizer, Takeda and Roche. M. Krebs declares honoraria from Roche, Jannsen, Octimer and Achilles Therapeutics. CM. Lovly has served as a consultant for Pfizer, Novartis, Astra Zeneca, Genoptix, Sequenom, ARIAD, Takeda, Foundation Medicine, Blueprints Medicine and Cepheid and has received research funding (to her university) from Novartis and Xcovery. The rest of the authors declare no potential conflicts of interest.

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
