## [Review Process File · EMBO Molecular Medicine]

Vulnerability of drug-resistant EML4-ALK rearranged lung cancer to transcriptional inhibition

Athanasios Rafail Paliouras, Lei Shi, Peter Magee, Sudhakar Sahoo, Hui Sun Leong, Matteo Fassan, Matthew Carter, Matthew Krebs, Fiona Blackhall, Christine Lovly, Michela Garofalo, Marta Buzzetti, Gianpiero Di Leva, and Ian Donaldson

DOI: [10.15252/emmm.201911099](https://doi.org/10.15252/emmm.201911099)

Corresponding author(s): Michela Garofalo (michela.garofalo@cruk.manchester.ac.uk)

Review Timeline:

Submission Date:	11th Jul 19
Editorial Decision:	20th Aug 19
Revision Received:	11th Dec 19
Editorial Decision:	23rd Jan 20
Revision Received:	23rd Apr 20
Editorial Decision:	12th May 20
Revision Received:	19th May 20
Accepted:	21st May 20

Editor: Lise Roth

Transaction Report:

20th Aug 2019

Dear Dr. Garofalo,

Thank you for the submission of your manuscript to EMBO Molecular Medicine. We have now heard back from the three referees whom we asked to evaluate your manuscript.

As you will see from the reports below, while they all mention the potential novelty and translational relevance of the findings, referees #1 and #2 also raise substantial and partly overlapping concerns, which should be convincingly addressed in a major revision of the present manuscript.

In particular, efforts should be made to:

- improve the logical cohesion of the manuscript,
- homogenize the cell lines used throughout the manuscript
- strengthen the mechanistic insights
- improve the in vivo part.

We realize that revising the manuscript according to the referees' recommendations will require a lot of additional work and experimentation. I am unsure whether you will be able or willing to address those and return a revised manuscript within the 3 months deadline. On the other hand, given the potential interest of the findings, I would be willing to consider a revised manuscript with the understanding that acceptance of the manuscript would entail a second round of review. EMBO Molecular Medicine encourages a single round of revision only and therefore, acceptance or rejection of the manuscript will depend on the completeness of your responses included in the next, final version of the manuscript. For this reason, and to save you from any frustration in the end, I would strongly advise against returning an incomplete revision and would also understand your decision if you choose to rather seek rapid publication elsewhere at this stage.

Should you find that the requested revisions are not feasible within the constraints outlined here and prefer, therefore, to submit your paper elsewhere, we would welcome a message to this effect.

I look forward to receiving your revised manuscript.

Yours sincerely,

Lise Roth

Lise Roth, PhD
Editor
EMBO Molecular Medicine

When submitting your revised manuscript, please carefully review the instructions that follow below. Failure to include requested items will delay the evaluation of your revision:

2) Individual production quality figure files as .eps, .tif, .jpg (one file per figure).

3) A .docx formatted letter INCLUDING the reviewers' reports and your detailed point-by-point responses to their comments. As part of the EMBO Press transparent editorial process, the point-by-point response is part of the Review Process File (RPF), which will be published alongside your paper.

4) A complete author checklist, which you can download from our author guidelines (<https://www.embopress.org/page/journal/17574684/authorguide#submissionofrevisions>). Please insert information in the checklist that is also reflected in the manuscript. The completed author checklist will also be part of the RPF.

6) Before submitting your revision, primary datasets produced in this study need to be deposited in an appropriate public database (see <https://www.embopress.org/page/journal/17574684/authorguide#dataavailability>). Please remember to provide a reviewer password if the datasets are not yet public. The accession numbers and database should be listed in a formal "Data Availability" section (placed after Materials & Method). Please note that the Data Availability Section is restricted to new primary data that are part of this study.

7) We would also encourage you to include the source data for figure panels that show essential data. Numerical data should be provided as individual .xls or .csv files (including a tab describing the data). For blots or microscopy, uncropped images should be submitted (using a zip archive if multiple images need to be supplied for one panel). Additional information on source data and instruction on how to label the files are available at

8) Our journal encourages inclusion of *data citations in the reference list* to directly cite datasets that were re-used and obtained from public databases. Data citations in the article text are distinct from normal bibliographical citations and should directly link to the database records from which the data can be accessed. In the main text, data citations are formatted as follows: "Data ref: Smith et al, 2001" or "Data ref: NCBI Sequence Read Archive PRJNA342805, 2017". In the Reference list, data citations must be labeled with "[DATASET]". A data reference must provide the database

name, accession number/identifiers and a resolvable link to the landing page from which the data can be accessed at the end of the reference. Further instructions are available at .

9) We replaced Supplementary Information with Expanded View (EV) Figures and Tables that are collapsible/expandable online. A maximum of 5 EV Figures can be typeset. EV Figures should be cited as 'Figure EV1, Figure EV2" etc... in the text and their respective legends should be included in the main text after the legends of regular figures.

- Additional Tables/Datasets should be labeled and referred to as Table EV1, Dataset EV1, etc. Legends have to be provided in a separate tab in case of .xls files. Alternatively, the legend can be supplied as a separate text file (README) and zipped together with the Table/Dataset file. See detailed instructions here:

10) The paper explained: EMBO Molecular Medicine articles are accompanied by a summary of the articles to emphasize the major findings in the paper and their medical implications for the non-specialist reader. Please provide a draft summary of your article highlighting

11) For more information: There is space at the end of each article to list relevant web links for further consultation by our readers. Could you identify some relevant ones and provide such information as well? Some examples are patient associations, relevant databases, OMIM/proteins/genes links, author's websites, etc...

12) Author contributions: the contribution of every author must be detailed in a separate section (before the acknowledgments).

13) A Conflict of Interest statement should be provided in the main text

14) Every published paper now includes a 'Synopsis' to further enhance discoverability. Synopses are displayed on the journal webpage and are freely accessible to all readers. They include a short stand first (maximum of 300 characters, including space) as well as 2-5 one-sentences bullet points that summarizes the paper. Please write the bullet points to summarize the key NEW findings. They should be designed to be complementary to the abstract - i.e. not repeat the same text. We encourage inclusion of key acronyms and quantitative information (maximum of 30 words / bullet point). Please use the passive voice. Please attach these in a separate file or send them by email, we will incorporate them accordingly.

Please also suggest a striking image or visual abstract to illustrate your article. If you do please provide a jpeg file 550 px-wide x 400-px high.

To submit your manuscript, please follow this link:

Link Not Available

***** Reviewer's comments *****

Referee #1 (Comments on Novelty/Model System for Author):

While individual experimental systems are adequate, their use through the paper is not always consistent.

Referee #1 (Remarks for Author):

Where the manuscript addresses an important clinical problem and represents a significant body of work, it lacks sufficient logical cohesion and the main conclusions appear to be somewhat contrived.

Specifically,

1. Use of multiple cell line models could have been a major strength of the study. However, instead of drawing inferences from a single model, and then testing for generalizability, authors use different model cell lines in different experiments without obvious reason or consistency. For example, the study starts with the analysis of transcriptome changes in STE1 cell line after a brief exposure to Crizotinib (Fig. 1 A-C). Then, it jumps to the analysis of miRNA expression in H3122 cell line without any justification, only to "validate" the results in STE1 cell line, focusing on just 2 miRNAs, 1 of which is not even shown in the Figure 1D. While Figure 1D shows clustering from CrizR1-R4, subsequent figures focus on R1, R4 & R5 - is there a specific reason for omitting R2&R3 and adding R5? R1 is used in all of the in vitro experiments, including those focusing on a single CrizR cell line, why was R4 chosen for the in vivo studies?

2. Given that ALK inhibitors have potent cytostatic effect, enrichment in the "cell cycle" category is trivial and to be expected, it is unclear how this finding connects with author's objective of "identifying transcriptomic changes that may mediate cell death response". Genes "driving" the category need to be listed in the supplement. It is also unclear why authors chose to focus exclusively on "cell death", rejecting other leads (such as Axl), as cytostatic effects of the drugs might be equally, or in some cases, more important.

3. It is unclear why authors prioritize focusing on miRNA, as the observed differences in expression are rather trivial (for example Fig. 1E); functional validation experiments with significant over-expression or deep suppression are not really addressing the impact of the more subtle changes observed in the resistant cell lines.

4. Demonstration of increased sensitivity of resistant cell lines to CDK inhibitors could be potentially of a significant clinical importance. However, the comparison needs to be made with therapy naive cells - and this critical reference point is lacking for many of the experiments. Conversely, I do not see a point of comparing responses of a cell line to a different drug - making statistical tests for significance in IC50 between crizotinib and CDK inhibitors - this is really apples to oranges

comparison, lacking in meaningful biological interpretability.

5. For the in vivo experiment with CrizR4 cell line, authors should have used parental cell line control to assess both crizotinib and alvocidib sensitivity.

6. Dramatic differences in % of apoptotic cells in xenograft tumors 2 weeks after discontinuation of the treatment are really puzzling. I am not sure the results are interpretable in the light of other data that authors are showing, as alvocidib should be completely out of circulation at this point, and any direct effects long gone. Also, the dramatic increase in cell death shown in Fig. 6C is highly inconsistent with lack of tumor shrinkage (Fig 6B) - I am not sure that the observations are reconcilable. Authors should provide more details on the IHC analysis, including the number of sections analyzed per tumor, and numbers of cells counted.

7. Authors should have performed a more detailed characterization of crizotinib resistant cell lines used in the study. How stable is the resistance in the absence of the treatment? What are the rates of proliferation and death of the cells in and out of the drugs?

Minor points:

1. Is there a rationale for not using curve fitting in Fig. 2B, in contrast to most of the rest figures showing dose responses?

2. Is there a reason for listing a full gene name for Vimentin, while using abbreviated names for other EMT related genes?

3. Unless crizotinib resistant cell lines were derived from single cells, they cannot be referred to as "clones".

4. I am confused about the "persisters" in Fig 4F - if the cells are capable of reaching confluence, they no longer fit into "persister" category. Also, it is unclear how the Ramirez et al reference relates to the 4F results.

Referee #2 (Comments on Novelty/Model System for Author):

CrizR1 is the cell line with better in vitro results. However, the xenograft model and treatment with Alvocidib was done with CrizR4, which generates some concerns. The in vivo experiment should be additionally done, at least, with CrizR1 cells. Also, I would find more accurate taking plasma samples from patients that are ALK wt (instead of mutant) but show resistance to the treatment.

I would give an opportunity to this manuscript, as I think the topic is interesting and clinically important. However, I have many concerns about the underlying mechanism (not clear at all) and the validation in mouse models and patient samples (weak). For the moment the article it is not suitable for the level and quality of EMBO Mol Med, most of the described results are negative, unless the authors substantially revise the manuscript, with more relevant and concluding results, and with a clearer rationale. It seems to me that they are taking different cell lines and a collection of drugs and performing a screening with all of the combinations, but a clear working hypothesis seems missing.

Referee #2 (Remarks for Author):

In this manuscript entitled "Vulnerability of drug-resistant EML4-ALK lung cancer cells to transcriptional inhibition" the authors present a comprehensive study on the molecular mechanisms underpinning response and resistance to ALK inhibition, and propose a new approach for the treatment of EML4-ALK lung cancer with acquired resistance to this type of inhibitors. This manuscript is well written and the authors put the findings into perspective considering published literature. While it poses an attractive strategy that could potentially be translated into the clinic for

this subset of patients, mechanistic data in the manuscript is at times insufficient to support the conclusions, which substantially affects the quality of the research.

Major concerns

1. The way the data is presented does not seem to orderly follow the hypothesis posed by the authors (whether miRNAs downstream of ALK signalling are pivotal to EMLK4-ALK cell survival and play a role in resistance to ALK inhibitors). miRNAs analysis is abandoned at the beginning of the study and different cues seem to be followed, only to forcibly make a new link at the end of the study with the mechanism proposed. While the text reads generally well, I believe the data is not generally presented in a logical manner as per the initial hypothesis and could be arranged in a more optimal manner, as well as panels in figures (data put together in the same figure, when they are independent "stories" is that makes sense?). For example, the first part of EGFR/TGFB in Figure 3 doesn't make any sense that is together with cell cycle dysregulation stuff. Figure 3 and 4 could be fused together as it comes from the same story, and many of the things could go to additional figures.
2. The mechanism in response to ALK inhibition is unclear. The statement that BIM is essential in this inhibition is largely unsupported. The only data shown on this analysis is protein levels, but authors do not show whether BIM function is affected or not, nor any data on cell survival/death upon BIM manipulation during treatment. Hence, follow-up during resistance is also unsupported.
3. The authors mention that parental H3122 is mutant for ALK, but the generated cell lines that are resistant for the drug. How do the authors explain that ALK is wildtype in resistant clones, if these come from a cell line that is essentially driven by mutant ALK?
4. Bembecentibid (Figure 2H-I) seems to be effectively decreasing cell viability in IC50 assay, but does not induce cell death. How do the authors explain these apparently contradictory results?
5. The transcriptional hypothesis as the driver of the effect seen by the inhibition of CDKs is largely weak. Authors do show in Figure EV4 a loss of phosphorylation upon treatment of CDK inhibitors, and then they show that KO of CDKs induced apoptosis, but they do not show definite data supporting this. Why didn't the authors check whether phosphorylation of RNA pol was rescued with KO of CDKs to test this hypothesis further?
Also, CDK6 silencing has no effect on the resistance of the cells to crizotinib, but overexpressing miRNAs that in turn decreases CDK6 does have effects? Do the authors have any additional explanation? Following my comments above it seems that the underlying mechanism driving this effect is not clear enough.
6. It is confusing why the authors show data on different cell lines depending on the experiment and what is the underlying rationale. CrizR1 is the one mostly shown in most in vitro analyses, however when authors move to the in vivo experiment they only use CrizR4, which has less effect in vitro than CrizR1. Why do they not show data on other cell lines?
7. Regarding the experiment with plasma of patients (Figure 7I), what about taking samples from patients that are ALK wt but show resistance to the treatment? Otherwise the results are not so relevant as the generated cell lines are resistant to the drug but are wt for the ALK locus. Alternatively, the authors should use cell lines with similar mutations than the human setting with resistance to the drug.

Minor points

In the same line that my comment on point 1, figures would need a reorganisation to highlight the essential and relevant data, and to move some material to the Supplementary Information, in particular negative results.

Referee #3 (Comments on Novelty/Model System for Author):

No issues

Referee #3 (Remarks for Author):

This revised manuscript is a well-structured study with merits of robust techniques and abundant data presentation. Authors aimed to explore the mechanisms of ALK inhibitor resistance, and hypothesized that some miRNAs downstream of ALK signalling are pivotal to EML4-ALK cell survival and could affect the response to crizotinib. Initially, they showed that response to ALK inhibition is mediated by a cell cycle dysregulation and BIM upregulation. They build up drug-resistant cell lines by long-term exposure to increasing concentration of ALK inhibitors and demonstrated EMT-related genes, particularly AXL, are upregulated in these resistant cells. They further disclosed cell cycle dysregulation in these resistant cells, which are remarkably sensitive to inhibition of CDK7/12 with THZ1 and CDK9 with alvociclib or dinaciclib. These compounds robustly induce apoptosis through transcriptional inhibition and downregulation of anti-apoptotic genes. Importantly, they demonstrated that alvociclib reduced tumour progression in an in vivo xenograft model. Finally, authors disclosed that miR-25 and miR-30c are upregulated in crizotinib-resistant cells and in plasma of patients who developed resistance in the clinic, therefore the development of a miRNA signature could serve as a diagnostic tool or biomarkers to predict the onset of resistance during treatment. These findings are novel and I think this manuscript is suitable for publish in this journal.

I have only one minor suggestion. Although authors described their experimental design and rationale very clearly and logically, their data and results are too robust to be understood easily. If authors could subtitle every section of results with brief conclusion, it would be very helpful for nonspecialist readers to comprehend the knowledge in this article.

Last, some words should be corrected in

1. WB "siSrambled" in Figure 5D and 5F should be "siScrambled"
2. In first line of Page 9, "encyclopaedia" should be "encyclopedia"

Point by point response to referees' comments

Referee #1 (Comments on Novelty/Model System for Author):

While individual experimental systems are adequate, their use through the paper is not always consistent.

Referee #1 (Remarks for Author):

Whereas the manuscript addresses an important clinical problem and represents a significant body of work, it lacks sufficient logical cohesion and the main conclusions appear to be somewhat contrived.

Specifically,

1. Use of multiple cell line models could have been a major strength of the study. However, instead of drawing inferences from a single model, and then testing for generalizability, authors use different model cell lines in different experiments without obvious reason or consistency. For example, the study starts with the analysis of transcriptome changes in STE1 cell line after a brief exposure to Crizotinib (Fig. 1 A-C). Then, it jumps to the analysis of miRNA expression in H3122 cell line without any justification, only to "validate" the results in STE1 cell line, focusing on just 2 miRNAs, 1 of which is not even shown in the Figure 1D. While Figure 1D shows clustering from CrizR1-R4, subsequent figures focus on R1, R4 & R5 - is there a specific reason for omitting R2&R3 and adding R5? R1 is used in all of the *in vitro* experiments, including those focusing on a single CrizR cell line, why was R4 chosen for the *in vivo* studies?

Authors' reply:

- We apologize for the confusion that the interchangeable use of EML4-ALK cell lines has caused. First, we would like to point out that we have now re-structured Figure 1 and removed miRNA data from the response to crizotinib that complicated, rather than aided the understanding of the central message of the manuscript.
- The CrizR4 cell line was chosen purely on technical grounds, based on its faster rate of growth *in vitro*. The fact that alvociclib had more modest effects on the CrizR4 cell line *in vitro* but still decent *in vivo* outcome, if anything, strengthens the findings. Despite this, we agree with the reviewer that this is a useful addition and a CrizR1 *in vivo* experiment has been added in Fig. 6B-C. Furthermore, we tested alvociclib in a xenograft mouse model of H3122 parental cells. This experiment is reported in Fig. 6A.

2. Given that ALK inhibitors have potent cytostatic effect, enrichment in the "cell cycle" category is trivial and to be expected, it is unclear how this finding connects with Authors' objective of "identifying transcriptomic changes that may mediate cell death response". Genes "driving" the category need to be listed in the supplement. It is also unclear why authors chose to focus exclusively on "cell death", rejecting other leads (such as Axl), as cytostatic effects of the drugs might be equally, or in some cases, more important.

Authors' reply:

- We agree with the reviewer that a cell cycle dysregulation is to be expected upon ALK inhibition. We did choose to validate the cell cycle pathway in a cell death context because it was the most significantly downregulated KEGG pathway and because cell cycle regulation

leads to cell death in some contexts. We have now re-written the text so that our main objective is more meaningfully communicated.

- The list of genes driving this category can be now found in (Appendix Dataset 2).
- We agree that a cytostatic effect for an anti-cancer compound is important. At the time we discovered an effect of AXL inhibition in ALK resistant cells we were in parallel evaluating CDK inhibitors, which as compared to AXL inhibition, had apoptotic effects. We thus decided to focus on the cytotoxic rather than only cytostatic compound. However, we have changed the text at page 5 which now reads: “In summary, we have detected an AXL-mediated induction of resistance to crizotinib. Although AXL inhibitors significantly reduce cell proliferation, they are unable to kill crizotinib resistant cells”.

3. It is unclear why authors prioritize focusing on miRNA, as the observed differences in expression are rather trivial (for example Fig. 1E); functional validation experiments with significant over-expression or deep suppression are not really addressing the impact of the more subtle changes observed in the resistant cell lines.

Authors' reply:

- We have now re-structured the manuscript and removed miRNA data on the response to crizotinib that complicated, rather than aided the understanding of the central message of the manuscript.

4. Demonstration of increased sensitivity of resistant cell lines to CDK inhibitors could be potentially of a significant clinical importance. However, the comparison needs to be made with therapy naive cells - and this critical reference point is lacking for many of the experiments. Conversely, I do not see a point of comparing responses of a cell line to a different drugs - making statistical tests for significance in IC50 between crizotinib and CDK inhibitors - this is really apple to oranges comparison, lacking in meaningful biological interpretability.

Authors' reply:

- We thank the reviewer for pointing out the suboptimal comparison of different compounds. We have now removed statistical comparisons and changed the figures to represent all the EML4-ALK cell lines treated separately with each inhibitor.
- All the proliferation experiments in the revised Figures 2 & 3 now represent drug-resistant cell lines compared with the drug-naïve, parental cell line and the same is true for the clonogenic assays in Figure 3. As indicated previously in our discussion, it is not surprising that the parental cells respond similarly to the drug-resistant cells, as the CDK-described vulnerability is inherent and not drug-induced.

5. For the *in vivo* experiment with CrizR4 cell line, authors should have used parental cell line control to assess both crizotinib and alvocidib sensitivity.

Authors' reply:

We have now expanded the *in vivo* experiments to include the parental H3122 and drug-resistant CrizR1 cells. These experiments can be found in Figure 6. Furthermore, response of AlecR cells to alvocidib *in vivo* has also been added in Figure 6E.

6. Dramatic differences in % of apoptotic cells in xenograft tumors 2 weeks after discontinuation of

the treatment are really puzzling. I am not sure the results are interpretable in the light of other data that authors are showing, as alvocidib should be completely out of circulation at this point, and any direct effects long gone. Also, the dramatic increase in cell death shown in Fig. 6C is highly inconsistent with lack of tumor shrinkage (Fig 6B) - I am not sure that the observations are reconcilable. Authors should provide more details on the IHC analysis, including the number of sections analyzed per tumor, and numbers of cells counted.

Authors' reply:

- We certainly agree that alvocidib is out of circulation at the time point of sacrifice and that does make interpretation of the IHC difficult. We speculate that some residual apoptosis did persist after the initial reduction of tumour progression. Furthermore, the quantification of IHC was actually only borderline significant. To avoid making unfounded claims for this, we have modified the text which now reads: "Upon sacrificing the mice, we assessed the apoptotic status of these tumours by *in situ* immunohistochemistry (IHC). The increase in cleaved caspase 3 was evident in CrizR1 and AlecR tumors (**Fig. 6F**) whilst in the CrizR4 tumors the levels of cleaved caspase 3 were marginally significant (**Appendix Fig S5E**). However, we should note that the absence of recent drug treatment in these mice complicates this interpretation".
- Furthermore, we added the results of two more xenograft mouse models using CrizR1 and AlecR cells injected subcutaneously in nude mice. Caspase 3 in these models is significantly increased in tumors treated with alvocidib as compared to vehicle control treated mice.
- More details regarding the IHC analysis have now been added in the 'Methods – IHC-quantification'. The text now reads "One section per tumour was imaged at 20x and the number of stain-positive cells was quantified in 5 random fields (One field contains approximately 1,600 cells). The mean number of positive cells was then plotted."

7. Authors should have performed a more detailed characterization of crizotinib resistant cell lines used in the study. How stable is the resistance in the absence of the treatment? What are the rates of proliferation and death of the cells in and out of the drugs?

Authors' reply:

- As suggested by the reviewer, we tested the proliferation of CrizR1, CrizR4 and AlecR cells kept in culture without crizotinib for up to 6 weeks. The cells showed no change in cell survival after crizotinib treatment for 72h as compared to cells constantly maintained in crizotinib.

Minor points:

1. Is there a rationale for not using curve fitting in Fig. 2B, in contrast to most of the rest figures showing dose responses?

Authors' reply:

- Thank you for pointing out this inconsistency, there is no rationale other than making the approximate IC_{50} value more easily distinguishable. This has now been changed to a fitted curve to ensure consistency with the other figures.

2. Is there a reason for listing a full gene name for Vimentin, while using abbreviated names for other EMT related genes?

Authors' reply:

- Thank you for pointing this out, Vimentin has now been changed to VIM.

3. Unless crizotinib resistant cell lines were derived from single cells, they cannot be referred to as "clones".

Authors' reply:

- This has now been changed to "cell lines" as indeed, there has not been single-cell cloning.

4. I am confused about the "persisters" in Fig 4F - if the cells are capable of reaching confluence, they no longer fit into "persister" category. Also, it is unclear how the Ramirez et al reference relates to the 4F results.

Authors' reply:

- Individual persister cells that survive an initial high drug dose will eventually reach confluence if grown for long enough periods (*Hangauer, Nature, 2017*). We apologize if this is not clear from our crystal violet staining, as cells tended to grow in large colonies and appear too confluent.
- The *Ramirez et al.*, reference was used to cite important findings about the concept of persister cells in oncogene-driven cancers (EGFR^{mut} lung cancer in this case). Also, we have now added the original reference, *Sharma et al., Cell, 2010* so that a non-familiar reader can find more information on the subject.

Referee #2 (Comments on Novelty/Model System for Author):

CrizR1 is the cell line with better in vitro results. However, the xenograft model and treatment with Alvocidib was done with CrizR4, which generates some concerns. The in vivo experiment should be additionally done, at least, with CrizR1 cells. Also, I would find more accurate taking plasma samples from patients that are ALK wt (instead of mutant) but show resistance to the treatment.

I would give an opportunity to this manuscript, as I think the topic is interesting and clinically important. However, I have many concerns about the underlying mechanism (not clear at all) and the validation in mouse models and patient samples (weak). For the moment the article it is not suitable for the level and quality of EMBO Mol Med, most of the described results are negative, unless the authors substantially revise the manuscript, with more relevant and concluding results, and with a clearer rationale. It seems to me that they are taking different cell lines and a collection of drugs and performing a screening with all of the combinations, but a clear working hypothesis seems missing.

Referee #2 (Remarks for Author):

In this manuscript entitled "Vulnerability of drug-resistant EML4-ALK lung cancer cells to transcriptional inhibition" the authors present a comprehensive study on the molecular mechanisms underpinning response and resistance to ALK inhibition, and propose a new approach for the treatment of EML4-ALK lung cancer with acquired resistance to this type of inhibitors. This manuscript is well written and the authors put the findings into perspective considering published literature. While it poses an attractive strategy that could potentially be translated into the clinic for

this subset of patients, mechanistic data in the manuscript is at times insufficient to support the conclusions, which substantially affects the quality of the research.

Major concerns

1. The way the data is presented does not seem to orderly follow the hypothesis posed by the authors (whether miRNAs downstream of ALK signalling are pivotal to EMLK4-ALK cell survival and play a role in resistance to ALK inhibitors). miRNAs analysis is abandoned at the beginning of the study and different cues seem to be followed, only to forcibly make a new link at the end of the study with the mechanism proposed. While the text reads generally well, I believe the data is not generally presented in a logical manner as per the initial hypothesis and could be arranged in a more optimal manner, as well as panels in figures (data put together in the same figure, when they are independent "stories" is that makes sense?). For example, the first part of EGFR/TGFB in Figure 3 doesn't make any sense that is together with cell cycle dysregulation stuff. Figure 3 and 4 could be fused together as it comes from the same story, and many of the things could go to additional figures.

Authors' reply:

- We have now re-arranged all the figures and text, separating the figures in a more logical fashion and moved much of negative results to supplementary figures.

2. The mechanism in response to ALK inhibition is unclear. The statement that BIM is essential in this inhibition is largely unsupported. The only data shown on this analysis is protein levels, but authors do not show whether BIM function is affected or not, nor any data on cell survival/death upon BIM manipulation during treatment. Hence, follow-up during resistance is also unsupported.

Authors' reply:

- We have now re-structured the manuscript and removed miRNA data from the response to crizotinib that complicated, rather than aided the understanding of the central message of the manuscript.

3. The authors mention that parental H3122 is mutant for ALK, but the generated cell lines that are resistant for the drug. How do the authors explain that ALK is wildtype in resistant clones, if these come from a cell line that is essentially driven by mutant ALK?

Authors' reply:

- We apologize for the confusion caused by describing the H3122 cells as being ALK mutant. In this, we meant that H3122 are mutant for ALK in the sense that they carry the EML4-ALK gene fusion. The same is true of course for the derived clones CrizR1, CrizR4, CrizR5 and CeritR which also have the EML4-ALK fusion. However, there is no ALK kinase domain mutation detected in these cells (either parental or resistant) that could change the structure and explain drug resistance through poor drug binding (i.e. L1196M mutation). We have now changed the text to read: "all the resistant cells have wild-type ALK kinase domain"

4. Bemcentinib (Figure 2H-I) seems to be effectively decreasing cell viability in IC50 assay, but does not induce cell death. How do the authors explain these apparently contradictory results?

Authors' reply:

- This effect on cell proliferation can be reconciled by a cytostatic instead of a cytotoxic effect. Indeed, several compounds have been described to produce cytostatic effects both *in vitro* and in the clinic, while not inducing cell death. For example, CDK4/6 inhibitors (Klein et al., *Cancer Cell*, 2018) and MEK inhibitors (Ruscetti et al., *Science*, 2018) are well-known to elicit cytostatic effects. While we have not examined this in our cells, a compatible explanation may be the induction of long-term senescence.

5. The transcriptional hypothesis as the driver of the effect seen by the inhibition of CDKs is largely weak. Authors do show in Figure EV4 a loss of phosphorylation upon treatment of CDK inhibitors, and then they show that KO of CDKs induced apoptosis, but they do not show definite data supporting this. Why didn't the authors check whether phosphorylation of RNA pol was rescued with KO of CDKs to test this hypothesis further? Also, CDK6 silencing has no effect on the resistance of the cells to crizotinib, but overexpressing miRNAs that in turn decreases CDK6 does have effects? Do the authors have any additional explanation? Following my comments above it seems that the underlying mechanism driving this effect is not clear enough.

Authors' reply:

- We have now performed more experiments to strengthen the mechanistic insights. We performed Chip-seq analysis after alvocidib and THZ1 treatment with RNA polymerase II pulldown. As seen in the new Figure 5, alvocidib treatment results in dramatic increase of the pausing of the RNA pol II at the TSS, highly suggestive of CDK9 inhibition, while THZ1 treatment results in decreased binding and recruitment of RNA pol II, suggestive of CDK7/CDK12 inhibition. This analysis has revealed that transcription of MYC and MYC targets along with other previously identified genes, namely CCND1, MCL-1 and Survivin, is inhibited by treatment with alvocidib or THZ1.
- The fact that CDK6 downregulation has no effect on cell viability while overexpression of miR-149 has pro-apoptotic effects suggests that miR-149 has other targets other than CDK6, which we have not discovered in this study, and whose downregulation is what causes cell death. Thus, we reason that the effect we observe upon miR-149 overexpression is CDK6-independent.

6. It is confusing why the authors show data on different cell lines depending on the experiment and what is the underlying rationale. CrizR1 is the one mostly shown in most in vitro analyses, however when authors move to the in vivo experiment they only use CrizR4, which has less effect in vitro than CrizR1. Why do they not show data on other cell lines?

Authors' reply:

- We regret that the use of different drug-resistant cell lines has been confusing and we hope that the major re-structuring of the figures and manuscript has helped.
- The CrizR4 cell line was chosen purely on technical grounds, based on its faster rate of growth in vitro. The fact that alvocidib had more modest effects on the CrizR4 cell line *in vitro* but still significant *in vivo* outcome, if anything, strengthens this finding. Despite this, we agree with the reviewer that this is a useful addition and we have now added in Fig. 6 the results of two more xenograft mouse models using CrizR1 and AlecR (Ste-1R) cells.

•

7. Regarding the experiment with plasma of patients (Figure 7I), what about taking samples from patients that are ALK wt but show resistance to the treatment? Otherwise the results are not so relevant as the generated cell lines are resistant to the drug but are wt for the ALK

locus. Alternatively, the authors should use cell lines with similar mutations than the human setting with resistance to the drug.

Authors' reply:

We apologize that our use of the term ALK^{mut} has caused confusion. We have now made clear that we are talking about patients that, as our cell lines, carry the EML4-ALK translocation (and therefore represent the patient population treated with ALK inhibitors). Figure 7I and the text have now been changed accordingly. Furthermore, we have indicated in a new table in Figure EV5C the patients that showed ALK-dependent mechanisms of resistance on progression. ALK has not been sequenced upon progression in all patients. Although 3 patients do have a mutation in the ALK kinase, as shown in Fig.EV5C, the fact that miR-25 and miR-30c are present in the blood of the majority of these patients when they progress suggests that they are prognostic for ALK patients that become resistant to crizotinib independently of the ALK status.

Minor points

In the same line that my comment on point 1, figures would need a reorganisation to highlight the essential and relevant data, and to move some material to the Supplementary Information, in particular negative results.

Authors' reply:

- Thank you for this suggestion, we have re-organized the text and figures in a more clear way.

Referee #3 (Comments on Novelty/Model System for Author):

No issues

Referee #3 (Remarks for Author):

This revised manuscript is a well-structured study with merits of robust techniques and abundant data presentation. Authors aimed to explore the mechanisms of ALK inhibitor resistance, and hypothesized that some miRNAs downstream of ALK signalling are pivotal to EML4-ALK cell survival and could affect the response to crizotinib. Initially, they showed that response to ALK inhibition is mediated by a cell cycle dysregulation and BIM upregulation. They build up drug-resistant cell lines by long-term exposure to increasing concentration of ALK inhibitors and demonstrated EMT-related genes, particularly AXL, are upregulated in these resistant cells. They further disclosed cell cycle dysregulation in these resistant cells, which are remarkably sensitive to inhibition of CDK7/12 with THZ1 and CDK9 with alvocidib or dinaciclib. These compounds robustly induce apoptosis through transcriptional inhibition and downregulation of anti-apoptotic genes. Importantly, they demonstrated that alvocidib reduced tumour progression in an in vivo xenograft model. Finally, authors disclosed that miR-25 and miR-30c are upregulated in crizotinib-resistant cells and in plasma of patients who developed resistance in the clinic, therefore the development of a miRNA signature could serve as a diagnostic tool or biomarkers to predict the onset of resistance during treatment. These findings are novel and I think this manuscript is suitable for publish in this journal.

I have only one minor suggestion. Although authors described their experimental design and rationale very clearly and logically, their data and results are too robust to be understood easily. If

authors could subtitle every section of results with brief conclusion, it would be very helpful for nonspecialist readers to comprehend the knowledge in this article.

Authors' reply:

- Thank you for the very positive feedback. We have now added a sentence at the end of each results section summarizing the key points.

Last, some words should be corrected in

1. WB "siSrambled" in Figure 5D and 5F should be "siScrambled"
2. In first line of Page 9, "encyclopaedia" should be "encyclopedia"

Authors' reply:

- Thank you very much for pointing out these mistakes, which have now been corrected.

23rd Jan 2020

Dear Prof. Garofalo,

Thank you for the submission of your revised manuscript to EMBO Molecular Medicine. We have now received feedback from the two reviewers who were asked to re-evaluate your manuscript. As you will see from the reports below, both referees acknowledge your efforts to address their initial concerns, and recognize that the manuscript has significantly improved. However, they also both mention issues that remain unanswered. While referee #1 states that minor revisions (without additional experiments) would be sufficient to bring the manuscript up to publication level, referee #2 finds additional *in vivo* experiments necessary to support the claims.

As EMBO Press encourages a single round of revisions only, we would normally reject the manuscript at this stage. However, as both reviewers recognize (as we do) the potential medical impact of the study and its interest for the community, we would like to exceptionally invite a second round of revisions. Please be aware that this will be the last chance for you to address the points raised by the referees.

When submitting your revised manuscript, please carefully review the instructions that follow below. Failure to include requested items will delay the evaluation of your revision:

2) Individual production quality figure files as .eps, .tif, .jpg (one file per figure).

3) A .docx formatted letter INCLUDING the reviewers' reports and your detailed point-by-point responses to their comments. As part of the EMBO Press transparent editorial process, the point-by-point response is part of the Review Process File (RPF), which will be published alongside your paper.

4) Before submitting your revision, primary datasets produced in this study need to be deposited in an appropriate public database (see <http://embomolmed.embopress.org/authorguide#dataavailability>).

The accession numbers and database should be listed in a formal "Data Availability" section (placed after Materials & Method). Please note that the Data Availability Section is restricted to new primary data that are part of this study.

5) We would also encourage you to include the source data for figure panels that show essential data. Numerical data should be provided as individual .xls or .csv files (including a tab describing the data). For blots or microscopy, uncropped images should be submitted (using a zip archive if multiple images need to be supplied for one panel). Additional information on source data and instruction on how to label the files are available at .

6) Our journal encourages inclusion of *data citations in the reference list* to directly cite datasets that were re-used and obtained from public databases. Data citations in the article text are distinct from normal bibliographical citations and should directly link to the database records from which the data can be accessed. In the main text, data citations are formatted as follows: "Data ref: Smith et al, 2001" or "Data ref: NCBI Sequence Read Archive PRJNA342805, 2017". In the Reference list, data citations must be labeled with "[DATASET]". A data reference must provide the database name, accession number/identifiers and a resolvable link to the landing page from which the data can be accessed at the end of the reference. Further instructions are available at .

7) Thank you for providing a synopsis image. Could you please upload it as a jpeg file 550 px-wide x 400-px high?

I look forward to receiving your revised manuscript.

Yours sincerely,

Lise Roth

Lise Roth, PhD
Editor
EMBO Molecular Medicine

To submit your manuscript, please follow this link:

Link Not Available

***** Reviewer's comments *****

Referee #1 (Comments on Novelty/Model System for Author):

The logical consistency of the paper could benefit from additional improvement, but it should be adequate for publication in the current form. Potential clinical and in vivo relevance would have been better served with the choice of models with strong initial responses to ALK inhibitors (unfortunately, H3122 tumors can grow in the presence of crizotinib in xenografts). Still, the data presented here is adequate and will be useful for the research community - I do not see a point in requesting additional experimental revisions.

Referee #1 (Remarks for Author):

The revision has addressed some of the key deficiencies of the original submission, though it still has a significant room for improvement. I do not want to delay the publication of this study. Instead, I suggest the authors to consider the following points, which could be addressed through a minor revision, without performing additional experiments.

1. Since the parental H3122 cells are sensitive to the CDK inhibitors, I think it needs to be reflected in the abstract - otherwise the abstract is somewhat misleading.
2. Given the sensitivity of the ALK inhibitor naive cells, I do not see much of a logical connection between the miR data and the rest of the manuscript. The paper essentially presents two unrelated stories. I suggest either separating the miR part into a different paper (with the potential biomarker emphasis), or figuring out how to better integrate it with the CDK inhibition part.
3. The authors did not resolve the confusion over the use of "persisters" term used to describe the data presented in 4F of the original submission (now Fig. 3B). This is not necessarily author's fault, as the universally accepted definitions of persisters are lacking, and there are no published studies explicitly examining changes in growth rates throughout the acquisition of resistance. Still, a statement "While persister cells treated only with crizotinib or alectinib maintained resistance ..." makes little sense to me. How is persistence different from resistance? Authors are incorrect in citing Hangauer Nature paper in support of their use of the term, as authors of that study associated persistence with lack of proliferation, and there, the "persisters" were allowed to reach confluence through drug removal, whereas in this manuscript cells were continuously grown under crizotinib. Likewise - there was nothing associating persistence with the ability to grow in the drug in Ramirez study - where resistance was contrasted with persistence, and the former was associated with stochastic acquisition of additional changes. Further, given that parental cells are also sensitive to 200 nM alvocidib, and since authors did not have a control of parental cells switched to adlocidib after reaching similar confluence as cells that started to grow in crizotinib, I am not sure what one can actually learn from this experiment - as it is unclear whether the cells that authors refer to as "persisters" are equally, more or less sensitive to the CDK inhibition compared to the parental or resistant cells. I think that authors should either dig deeper into this, or omit 3B as uninformative. Should authors decide to keep the figure, the time it took H3122 cells to start growing needs to be indicated.
4. It would have been useful to plot growth rates of H3122 cells and CrizR variants in and out of crizotinib on the same plot, perhaps in a supplementary figure.
5. It would be useful to have a color version of 3D and 4G - other panels have color anyway, and it will be easier to eyeball the relevant groups.
6. I do not see a point in having media and driving oncogene in 1A, as there is no variability between the cell lines there.
7. Consider putting a concluding sentence or two into the first paragraph in page 9 - to summarize inferences from the expression dataset analyses, and to have more general conclusion for the whole sub-section.

Referee #2 (Comments on Novelty/Model System for Author):

I think that the models, either in vitro and in vivo, are adequate, and that the overall technical quality for the manuscript is high, with state-of-the-art techniques including RNAseq, xenografts, IC50s and studies on miRNAs.

Regarding the novelty, I still think the paper lacks of cohesion and integrity in some parts, it looks like a compilation of results rather than a project based on a good rationale and working hypothesis.

You can clearly notice when you read the paper how the different sections are sometimes misconnected in the text of the Results. Sometimes it looks very forced (e.g. "Considering the efficacy of both alvocidib and dinaciclib in inhibiting the transcriptional CDK9,..." Why CDK9 but not other CDKs? It seems that the authors have tested and screened a set of drugs and tried to build a story based on positive results, but not always with a clear rationale behind.

As per the medical impact, I think the findings are compromised by an uncompleted *in vivo* validation. I appreciate a lot the efforts of the authors, as they have improved the paper a lot with further insights in the mechanism and with tumour xenografts. However, I find crucial that, at least some of these claims, are sustained by the *in vivo* validation. Please find my specific comments on the validation of the transcriptional regulators, the use of Lorlatinib, or the miRNAs. These findings are not validated in the *in vivo* context.

Despite these weaknesses, I still think that with a little bit more of work the article may be suitable for EMBO Mol Med. As I said, the authors did an intensive work, and targeting resistant cells is very challenging. In this sense I think that, with a proper *in vivo* validation, the medical impact could be definitively higher.

Referee #2 (Remarks for Author):

This is the revised version of the manuscript entitled "Vulnerability of drug-resistant EML4-ALK rearranged lung cancer to transcriptional inhibition by Paliouras et al. The authors show that EML4-ALK cells resistant to crizotinib, ceritinib or alectinib are sensitive to inhibition of CDK7/12 with THZ1, and CDK9 with alvocidib and dinaciclib, and provide additional information on the mechanisms of resistance of lung adenocarcinoma cells to ALK inhibitors. This includes evidence that some CDK inhibitors induce apoptosis through transcriptional inhibition of MYC and MYC targets. Of note, the authors have expanded the *in vivo* experiments with xenografts and provide further data on the role of miRNAs in the regulation of cell cycle genes involved in crizotinib resistance.

Despite the fact the manuscript has been substantially improved by the authors I still think it often lacks of logical cohesion and the rationale of the experiments seems sometimes not obvious or forced. The revised article has addressed some but not all my raised concerns. I am still worried about some of the conclusions and the potential applications of this work in clinical settings for the reasons stated below:

Figure 1. Bemcentinib does not have an effect in cell proliferation alone but only in combination with Crizotinib. How can the authors explain this? I think a possible cytostatic effect or the induction of cellular senescence should have been tested as an internal control along the different treatments used in the disclosed figures. It is well known that some CDK inhibitors can induce cellular senescence instead of, or simultaneously to, apoptosis (e.g. palbociclib).

Figures 2 and 3. It is difficult to evaluate to what extent alvocidib, dinaciclib and THZ1 inhibits cell proliferation in EML4-ALK cells when compared to the parental cells (e.g. Figures 2G, 2H and 3A). The differences are not significant and the inhibitors seem to affect both parental and CriR resistant cells in a similar manner. This result tempers the enthusiasm for drugs or inhibitors specifically killing or inducing apoptosis in the resistant cells.

Figures 4 and 5. The paper was notably improved in the new Figures 4 and 5. I wonder whether, in line with my comments below, it would be possible some *in vivo* validation. Is there a correlation of

tumour progression and the expression of MYC and MYC targets in the xenograft experiments of Figure 6? The same applies to Survivin, MCL-1 and BIM. At least a validation by qRT-PCR in snap frozen tumors would be relevant and strongly strengthen the manuscript.

In addition to MYC targets, KRas signalling was also found in the GSEA analysis of Figure 5B. I am just curious if the authors have tried to inhibit the Ras pathway, for example with a Mek inhibitor; or the PI3Kinase pathway, for instance by using rapamycin or BEZ235, in order to validate the role of MYC (although I am aware that targeting MYC may be challenging).

Figure 6. This is one of the most important figures in the study. I appreciate the efforts of the authors to include new data with CrizR1 and AlecR xenografts. However, I continue to be worried about how robust the finding is, and how relevant is the inhibition of tumour growth. If the model proposed by the authors in Figure 6H is correct, I find crucial to show that lorlatinib has no impact in CrizR1, Criz4 or AlecR xenografts and not only with cells in culture (as shown in Figure 6G). If it were the case, then there would be a clearer translatability of the claims.

Another concern is that the N used is not stated in the Figure Legend, except for Figure 6D, but according to the tumour weight graphs the numbers of tumors range from 4 to 6 (Figure 6C and 6E), so it is difficult to draw solid conclusions, especially considering the variability in tumor size and the slow rate of growth. The experiment would also benefit from a more detailed histological analysis, for instance by adding markers of proliferation (e.g. ki67) and cell cycle-inhibitors (p21, p16, etc.) and not only by using the apoptosis marker. The proposed independence of cell cycle arrest in the model of action of the compounds should be correlated with this analysis.

Regarding the potential translatability, could the authors comment or show data on the potential toxicities of the treatments in mice?

Figure 7. Although interesting, I think this part seems a Figure disconnected with the rest of the manuscript, as it is more focused on prognosis and potential biomarkers. As mentioned in text, some of the miRNAs were previously published by the authors (such as miR-25 and miR-30c). If tested and validated *in vivo*, it would be much more relevant. For instance, by analyzing the miRNAs from the serum of the mice used in the previous Figure 6, and to correlate them with the human samples. Have the authors collected blood samples from their mice to this end? The absence of this data and validations exemplifies the lack of cohesion observed in some parts of the manuscript.

Point by point response to referees' comments (2nd revisions)

Referee #1 (Comments on Novelty/Model System for Author):

The logical consistency of the paper could benefit from additional improvement, but it should be adequate for publication in the current form. Potential clinical and in vivo relevance would have been better served with the choice of models with strong initial responses to ALK inhibitors (unfortunately, H3122 tumors can grow in the presence of crizotinib in xenografts). Still, the data presented here is adequate and will be useful for the research community - I do not see a point in requesting additional experimental revisions.

Referee #1 (Remarks for Author):

The revision has addressed some of the key deficiencies of the original submission, though it still has a significant room for improvement. I do not want to delay the publication of this study. Instead, I suggest the authors to consider the following points, which could be addressed through a minor revision, without performing additional experiments.

1. Since the parental H3122 cells are sensitive to the CDK inhibitors, I think it needs to be reflected in the abstract - otherwise the abstract is somewhat misleading.

Authors' reply:

We would like to thank the reviewer for pointing this out. We have added this information in the abstract, which now reads: "Exploring these mechanisms of resistance, we found that EML4-ALK cells parental or resistant to crizotinib, ceritinib or alectinib are remarkably sensitive to inhibition of CDK7/12 with THZ1 and CDK9 with alvocidib or dinaciclib".

2. Given the sensitivity of the ALK inhibitor naive cells, I do not see much of a logical connection between the miR data and the rest of the manuscript. The paper essentially presents two unrelated stories. I suggest either separating the miR part into a different paper (with the potential biomarker emphasis), or figuring out how to better integrate it with the CDK inhibition part.

Authors' reply:

Following the reviewer's suggestion, we have now deleted miRNA-related data.

3. The authors did not resolve the confusion over the use of "persisters" term used to describe the data presented in 4F of the original submission (now Fig. 3B). This is not necessarily author's fault, as the universally accepted definitions of persisters are lacking, and there are no published studies explicitly examining changes in growth rates throughout the acquisition of resistance. Still, a statement "While persister cells treated only with crizotinib or alectinib maintained resistance ..." makes little sense to me. How is persistence different from resistance? Authors are incorrect in citing Hangauer Nature paper in support of their use of the term, as authors of that study associated persistence with lack of proliferation, and there, the "persisters" were allowed to reach confluence through drug removal, whereas in this manuscript cells were continuously grown under crizotinib. Likewise - there was nothing associating persistence with the ability to grow in the drug in Ramirez study - where resistance was contrasted with persistence, and the former was associated with stochastic acquisition of additional changes. Further, given that parental cells are also sensitive to 200 nM alvocidib, and since authors did not have a control of parental cells switched to adlocidib after reaching similar confluence as cells that started to grow in crizotinib, I am not sure what one can actually learn from this experiment - as it is unclear whether the cells that authors refer to as "persisters" are equally, more or less sensitive to the CDK inhibition compared to the parental or resistant cells. I think that authors should either dig deeper into this, or omit 3B as uninformative. Should authors decide to keep the figure, the time it took H3122 cells to start growing needs to be indicated.

Authors' reply:

- We have now deleted Fig. 3B.

4. It would have been useful to plot growth rates of H3122 cells and CrizR variants in and out of crizotinib on the same plot, perhaps in a supplementary figure.

Authors' reply:

- We have added a single plot with growth rates of H3122 cells and CrizR variants in Fig. EV4A.

5. It would be useful to have a color version of 3D and 4G - other panels have color anyway, and it will be easier to eyeball the relevant groups.

Authors' reply:

- We thank the reviewer for pointing this out. We have now added a color version of Figs. 3C, 4G and 5G.

6. I do not see a point in having media and driving oncogene in 1A, as there is no variability between the cell lines there.

Authors' reply:

- We have modified the table in Fig. 1A accordingly.

7. Consider putting a concluding sentence or two into the first paragraph in page 9 - to summarize inferences from the expression dataset analyses, and to have more general conclusion for the whole sub-section.

Authors' reply:

- We have addressed this point.

Referee #2 (Comments on Novelty/Model System for Author):

I think that the models, either in vitro and in vivo, are adequate, and that the overall technical quality for the manuscript is high, with state-of-the-art techniques including RNAseq, xenografts, IC50s and studies on miRNAs.

Regarding the novelty, I still think the paper lacks of cohesion and integrity in some parts, it looks like a compilation of results rather than a project based on a good rationale and working hypothesis. You can clearly notice when you read the paper how the different sections are sometimes misconnected in the text of the Results. Sometimes it looks very forced (e.g. "Considering the efficacy of both alvocidib and dinaciclib in inhibiting the transcriptional CDK9,..." Why CDK9 but not other CDKs? It seems that the authors have tested and screened a set of drugs and tried to build a story based on positive results, but not always with a clear rationale behind.

Authors' reply:

- We deleted data on the involvement of microRNAs in response to crizotinib, which was clearly disconnected from the main results of the manuscript. For clarity, we changed the mentioned sentence at page 5, which now reads: "Considering the efficacy of both alvocidib and dinaciclib in inhibiting the transcriptional CDKs and the preferential activity against CDK9".

As per the medical impact, I think the findings are compromised by an uncompleted in vivo validation. I appreciate a lot the efforts of the authors, as they have improved the paper a lot with further insights in the mechanism and with tumour xenografts. However, I find

crucial that, at least some of these claims, are sustained by the in vivo validation. Please find my specific comments on the validation of the transcriptional regulators, the use of Lorlatinib, or the miRNAs. These findings are not validated in the in vivo context.

Despite these weaknesses, I still think that with a little bit more of work the article may be suitable for EMBO Mol Med. As I said, the authors did an intensive work, and targeting resistant cells is very challenging. In this sense I think that, with a proper in vivo validation, the medical impact could be definitively higher.

Referee #2 (Remarks for Author):

This is the revised version of the manuscript entitled "Vulnerability of drug-resistant EML4-ALK rearranged lung cancer to transcriptional inhibition by Paliouras et al. The authors show that EML4-ALK cells resistant to crizotinib, ceritinib or alectinib are sensitive to inhibition of CDK7/12 with THZ1, and CDK9 with alvociclib and dinaciclib, and provide additional information on the mechanisms of resistance of lung adenocarcinoma cells to ALK inhibitors. This includes evidence that some CDK inhibitors induce apoptosis through transcriptional inhibition of MYC and MYC targets. Of note, the authors have expanded the in vivo experiments with xenografts and provide further data on the role of miRNAs in the regulation of cell cycle genes involved in crizotinib resistance.

Despite the fact the manuscript has been substantially improved by the authors I still think it often lacks of logical cohesion and the rationale of the experiments seems sometimes not obvious or forced. The revised article has addressed some but not all my raised concerns. I am still worried about some of the conclusions and the potential applications of this work in clinical settings for the reasons stated below:

Figure 1. Bemcentinib does not have an effect in cell proliferation alone but only in combination with Crizotinib. How can the authors explain this? I think a possible cytostatic effect or the induction of cellular senescence should have been tested as an internal control along the different treatments used in the disclosed figures. It is well known that some CDK inhibitors can induce cellular senescence instead of, or simultaneously to, apoptosis (e.g. palbociclib).

Authors' reply:

- We thank the reviewer for the very positive and constructive comments. We sympathise with the reviewer's concern as we tried to answer the same question. Bemcentinib treatment alone or in combination with crizotinib for 72h did not induce senescence as compared to cells treated with palbociclib. This experiment has been added in Appendix Fig. S2B. Bemcentinib treatment in combination with crizotinib has a cytostatic effect, as no apoptosis was observed. It is well known that acquired

resistance originates from the activation of parallel compensatory mechanisms. We believe that in crizotinib resistant cells oncogenic pathways such as the ERK/MEK pathway could be activated via AXL upregulation. Therefore, a combination treatment of crizotinib plus bemcentinib sensitizes cells to crizotinib, although does not induce apoptosis. Furthermore, AXL inhibition via bemcentinib has been previously shown to re-establish drug sensitivity in many models of acquired resistance, including erlotinib-resistant head and neck cancer cells (*Brand TM et al., clinical Cancer Res, 21 (11), 2601-12*); *Zhu C et al., PMID: 31684958*). Senescence could not be addressed upon alvocidib or THZ1 treatment, as these drugs induce massive and rapid cell death.

Figures 2 and 3. It is difficult to evaluate to what extent alvocidib, dinaciclib and THZ1 inhibits cell proliferation in EML4-ALK cells when compared to the parental cells (e.g. Figures 2G, 2H and 3A). The differences are not significant and the inhibitors seem to affect both parental and CriR resistant cells in a similar manner. This result tempers the enthusiasm for drugs or inhibitors specifically killing or inducing apoptosis in the resistant cells.

Authors' reply:

- Although it is true that CDK inhibitors are effective in the resistant as well in the parental ALK-EML4 cells, we do not see why this should temper the enthusiasm for these inhibitors. We have demonstrated that ALK/EML4 cells either parental or with acquired resistance to ALK inhibitors, have an inherent vulnerability to CDK inhibitors. In terms of sequential use, we propose that CDK inhibitors could be an effective strategy for patients with wild-type ALK after ALK inhibitors (first, second or third generation) have failed.

Figures 4 and 5. The paper was notably improved in the new Figures 4 and 5. I wonder whether, in line with my comments below, it would be possible some in vivo validation. Is there a correlation of tumour progression and the expression of MYC and MYC targets in the xenograft experiments of Figure 6? The same applies to Survivin, MCL-1 and BIM. At least a validation by qRT-PCR in snap frozen tumors would be relevant and strongly strengthen the manuscript.

Authors' reply:

- We have addressed this point in Fig. 6E, Fig. 6F and in EV5C. MYC, p21, survivin and MCL-1 have been analysed in the xenograft mouse models reported in this study by qRT-PCR or IHC.

In addition to MYC targets, KRas signalling was also found in the GSEA analysis of Figure 5B. I am just curious if the authors have tried to inhibit the Ras pathway, for example with a Mek

inhibitor; or the PI3Kinase pathway, for instance by using rapamycin or BEZ235, in order to validate the role of MYC (although I am aware that targeting MYC may be challenging).

Authors' reply:

- We thank the reviewer for pointing this out. We have inhibited the RAS pathway with trametinib, the PI3K pathway with rapamycin or both with a drug combination and observed an induction of cell death with the drug alone or in combination. This experiment has been added in Fig. 5H.

Figure 6. This is one of the most important figures in the study. I appreciate the efforts of the authors to include new data with CrizR1 and AlecR xenografts. However, I continue to be worried about how robust the finding is, and how relevant is the inhibition of tumour growth.

We have shown that the concentration of alvocidib we used *in vivo* (10mg/kg) drastically induces p21 and cleaved caspase 3 and reduces Ki67 expression levels (Fig.6D, Fig.6 E and Fig. EV5B). It is likely that an increase in drug concentration would further reduce tumor growth.

If the model proposed by the authors in Figure 6H is correct, I find crucial to show that lorlatinib has no impact in CrizR1, Criz4 or AlecR xenografts and not only with cells in culture (as shown in Figure 6G). If it were the case, then there would be a clearer translatability of the claims.

Authors' reply:

As suggested by the reviewer, we performed an *in vivo* study by injecting the CrizR1 cells subcutaneously in NSG mice and then mice were treated with lorlatinib (5 mg/kg) or vehicle control. Although cells with acquired resistance to ALK inhibitors were resistant to different concentrations of lorlatinib *in vitro* up to 300 μ M (Fig. 6G and data not shown), CrizR1 tumors were sensitive to lorlatinib *in vivo*. This result could be due to the different concentrations of lorlatinib used *in vitro* and *in vivo*. It is possible that in the *in vivo* study we exceeded the concentration to which CrizR1 cells are resistant to the drug. Nevertheless, we believe that this experiment adds knowledge on the response of crizotinib-resistant ALK wild-type tumors to lorlatinib and it has been added in Fig.6H. In addition, we revised the model in Fig. 6I. Lorlatinib could be used as second line therapy in patients with wild-type ALK who became refractory to crizotinib, while alvocidib could be used when resistance to ALKi occurs as potential alternative to chemotherapy.

Another concern is that the N used is not stated in the Figure Legend, except for Figure 6D, but according to the tumour weight graphs the numbers of tumors range from 4 to 6 (Figure 6C and 6E), so it is difficult to draw solid conclusions, especially considering the variability in tumor size and the slow rate of growth.

Authors' reply:

- We have now added the number of mice in the Figure legends.

The experiment would also benefit from a more detailed histological analysis, for instance by adding markers of proliferation (e.g. ki67) and cell cycle-inhibitors (p21, p16, etc.) and not only by using the apoptosis marker. The proposed independence of cell cycle arrest in the model of action of the compounds should be correlated with this analysis.

Authors' reply:

- We have analysed Ki67 and p21 by IHC or qPCR. Data has been added in Fig.6E and EV5B.

Regarding the potential translatability, could the authors comment or show data on the potential toxicities of the treatments in mice?

Authors' reply:

Although some mice treated with alvocidib showed weight loss compared to mice treated with vehicle, this does not hold true for all the studies. In the AlecR experiment, 3 out of 7 mice were sacrificed because of poor health conditions (of which one for weight loss). However, this experiment lasted longer (52 days) compared to the other studies (15 days).

Below we report the final and initial number of mice in each study:

H3122: vehicle=5/5, alvocidib=4/8 (of which 2 mice had to be sacrificed for tumour ulceration)

CR1: vehicle=4/5, alvocidib=6/8

CR4: vehicle= 5/7, alvocidib=7/7

AlecR: vehicle=6/7, alvocidib=4/7

It should be kept in consideration that the mouse strain used in these studies, NOD-SCID mice, is an immunodeficient mouse strain. In addition, alvocidib is or has been tested in 66 different clinical trials for patients with hematologic or solid malignancies (<https://clinicaltrials.gov/ct2/results?cond=&term=alvocidib&cntry=&state=&city=&dist=>) and it has been shown to be relatively well tolerated (*Senderowicz AM, et al., Phase I Trial of*

Continuous Infusion Flavopiridol, a Novel Cyclin-Dependent Kinase Inhibitor, in Patients With Refractory Neoplasms J Clin Oncol 1998 Sep;16(9):2986-99).

Figure 7. Although interesting, I think this part seems a Figure disconnected with the rest of the manuscript, as it is more focused on prognosis and potential biomarkers. As mentioned in text, some of the miRNAs were previously published by the authors (such as miR-25 and miR-30c). If tested and validated in vivo, it would be much more relevant. For instance, by analyzing the miRNAs from the serum of the mice used in the previous Figure 6, and to correlate them with the human samples. Have the authors collected blood samples from their mice to this end? The absence of this data and validations exemplifies the lack of cohesion observed in some parts of the manuscript.

Authors' reply:

- We agree with the reviewer and we have now deleted miRNA-related data.

12th May 2020

Dear Dr. Garofalo,

Thank you for the submission of your revised manuscript to EMBO Molecular Medicine. We have now received the enclosed report from referee #2, who reviewed the new version of your manuscript. As you will see, this referee is now supportive of publication, and I am thus pleased to inform you that we will be able to accept your manuscript pending the following final editorial amendments:

1) Please address referee #2's minor comment regarding toxicities of the tested compounds.

2) Main manuscript text:

- Please correct/answer the track changes suggested by our data editors in the main manuscript file (in track changes mode).
- Please remove the highlighted text.
- Please provide up to 5 keywords.
- Remove "data not shown". As per our guidelines, on "Unpublished Data" the journal does not permit citation of "Data not shown". All data referred to in the paper should be displayed in the main or Expanded View figures. "Unpublished observations" may be referred to in exceptional cases, where these are data peripheral to the major message of the paper and are intended to form part of a future or separate study, the names of the persons that reported the observation should be listed in brackets. Personal communications (Author name(s), personal communications) must be authorised in writing by those involved, and the authorisation sent to the editorial office at time of submission.
- Please indicate in legends or in the figures the exact $n=$ and exact $p=$ values, not a range, along with the statistical test used. Some people found that to keep the figures clear, providing a supplemental table with all exact p -values was preferable. You are welcome to do this if you want to.
- In the Material and Methods section, the paragraph on patients' samples has been removed; please clarify how the samples were collected and their use. Please also include a statement that informed consent was obtained from all subjects and that the experiments conformed to the principles set out in the WMA Declaration of Helsinki and the Department of Health and Human Services Belmont Report. This should also be stated in the checklist.

3) Figures:

- Please rename the legends to have "Appendix Figure S1" etc.
- Please remove the references to appendix table and dataset from the appendix table of content, and rename them "Table EV1" and "Dataset EV1" etc. in the files, legends and callouts (references in the main manuscript text).

4) Source data:

Thank you for providing source data for your figures. Please upload them as one PDF file per figure.

5) Checklist:

Part F/20: you indicated "Access has been provided", but this part is only meant for human genomic datasets. Please clarify.

6) Data availability section: Thank you for depositing your data in the Array Express repository. Please rephrase following the example below:

"The datasets produced in this study are available in the following databases:

- [data type]: [full name of the resource] [accession number/identifier] ([doi or URL or identifiers.org/DATABASE:ACCESSION])

Examples:

- RNA-Seq data: Gene Expression Omnibus GSExxxxx
(<https://www.ncbi.nlm.nih.gov/geo/query/acc.cgi?acc=GSExxxxx>)
- Chip-Seq data: Gene Expression Omnibus GSEyyyyy
(<https://www.ncbi.nlm.nih.gov/geo/query/acc.cgi?acc=GSEyyyyy>)"

Please note that the data under the code E-MTAB-8380 is not yet accessible to the public, which should be resolved before acceptance.

7) As part of the EMBO Publications transparent editorial process initiative (see our Editorial at <http://embomolmed.embopress.org/content/2/9/329>), EMBO Molecular Medicine will publish online a Review Process File (RPF) to accompany accepted manuscripts.

This file will be published in conjunction with your paper and will include the anonymous referee reports, your point-by-point response and all pertinent correspondence relating to the manuscript. Let us know whether you agree with the publication of the RPF and as here.

I look forward to receiving your revised manuscript.

Yours sincerely,

Lise Roth

Lise Roth, PhD
Editor
EMBO Molecular Medicine

To submit your manuscript, please follow this link:

Link Not Available

The system will prompt you to fill in your funding and payment information. This will allow Wiley to

send you a quote for the article processing charge (APC) in case of acceptance. This quote takes into account any reduction or fee waivers that you may be eligible for. Authors do not need to pay any fees before their manuscript is accepted and transferred to our publisher.

***** Reviewer's comments *****

Referee #2 (Comments on Novelty/Model System for Author):

NSCLC is a devastating disease and this paper has a potential clinical dimension in the treatment of patients with acquired resistance to ALK inhibitors. This is a high quality work including state-of-the-art techniques including RNAseq, cell proliferation and viability assays, molecular biology techniques and in vivo models. Overall, I found adequate the in vitro and in vivo models used.

Referee #2 (Remarks for Author):

I am satisfied with the author's comments and additional experiments, as they have addressed most of my original concerns. My only suggestion would be to briefly include a comment in the main text on the absence of relevant toxicities in the treatments in vivo, and any data supporting this conclusion in the supplementary material (e.g. weights of the mice, see also the reply of the authors to my question), as I think it would be relevant.

I congratulate the authors for such a meticulous work.

The authors performed the requested changes.

21st May 2020

Dear Dr. Garofalo,

Thank you for submitting your revised version of the manuscript and the new checklist.
I am now very pleased to accept your manuscript for publication in EMBO Molecular Medicine!

Congratulations on your interesting work,

Sincerely,

Lise Roth

Lise Roth, Ph.D
Editor
EMBO Molecular Medicine

Follow us on Twitter @EmboMolMed
Sign up for eTOCs at embopress.org/alertfeeds

Corresponding Author Name: Michela Garofalo

Manuscript Number: EMM-2019-11099